# Right inferior frontal gyrus implements motor inhibitory control via beta-band oscillations in humans

Michael Schaum[1]*, Edoardo Pinzuti[1], Alexandra Sebastian[2], Klaus Lieb[2], Pascal Fries[3,4], Arian Mobascher[2], Patrick Jung[2], Michael Wibral[5†], Oliver Tüscher[2†]

[1]Systemic Mechanisms of Resilience, Leibniz Institute for Resilience Research, Mainz, Germany; [2]Department of Psychiatry and Psychotherapy, University Medical Center of the Johannes Gutenberg University, Mainz, Germany; [3]Ernst Strüngmann Institute (ESI) for Neuroscience in Cooperation with Max Planck Society, Frankfurt, Germany; [4]Donders Institute for Brain, Cognition and Behaviour, Radboud University Nijmegen, Nijmegen, Netherlands; [5]Campus Institute Dynamics of Biological Networks, Georg-August University, Göttingen, Germany

**Abstract** Motor inhibitory control implemented as response inhibition is an essential cognitive function required to dynamically adapt to rapidly changing environments. Despite over a decade of research on the neural mechanisms of response inhibition, it remains unclear, how exactly response inhibition is initiated and implemented. Using a multimodal MEG/fMRI approach in 59 subjects, our results reliably reveal that response inhibition is initiated by the right inferior frontal gyrus (rIFG) as a form of attention-independent top-down control that involves the modulation of beta-band activity. Furthermore, stopping performance was predicted by beta-band power, and beta-band connectivity was directed from rIFG to pre-supplementary motor area (pre-SMA), indicating rIFG's dominance over pre-SMA. Thus, these results strongly support the hypothesis that rIFG initiates stopping, implemented by beta-band oscillations with potential to open up new ways of spatially localized oscillation-based interventions.

**\*For correspondence:**
Michael.Schaum@lir-mainz.de

†These authors contributed equally to this work

**Competing interests:** The authors declare that no competing interests exist.

## Introduction

Response inhibition is an essential component of cognitive control. Many neuropsychiatric disorders like attention deficit hyperactivity disorder, obsessive-compulsive disorder, and Parkinson's disease demonstrate the severe impact of impaired response inhibition on mental health (*Schachar and Logan, 1990*; *de Wit et al., 2012*; *Voon and Dalley, 2011*). Response inhibition has been operationalized via the classical stop-signal task (SST, *Logan et al., 1984*) in many studies; thus the SST plays a major role in defining research on cognitive control (*Verbruggen et al., 2019*). In the 'GO' condition of this task, participants have to respond to a GO signal by executing an immediate motor response, for example, performing a button press. In the less frequent 'STOP' condition, the GO signal is followed by a STOP signal after a variable delay and participants have to cancel the planned or ongoing motor response. A well-developed theoretical framework describing the cognitive processes underlying response inhibition relies on a so-called independent race model, which assumes the independence of go and stop processes (*Logan et al., 1984*; *Verbruggen et al., 2019*). Depending on which process finishes first, the stop process, which is initiated after the variable stop-signal delay, or the go process, which is initiated after the GO signal, the response can be inhibited successfully or not. Based on this framework, the latency of the unobservable stop response, the

stop-signal reaction time (SSRT), can be estimated by subtracting the mean stop-signal delay (SSD) from the mean GO RT (in the simplest way).

So far, numerous studies using the SST provided consistent evidence that response inhibition crucially depends on two cortical regions: the inferior frontal cortex (IFC) and the pre-supplementary motor area (pre-SMA) (*Wessel and Aron, 2017*). The particular functional roles of these two regions within response inhibition have been controversially debated for more than a decade now (*Aron et al., 2004*; *Hampshire and Sharp, 2015a*; *Aron et al., 2015*; *Hampshire and Sharp, 2015b*). While a range of studies on response inhibition point toward a single inhibitory module within the IFC that directly initiates stopping (*Aron et al., 2003*), other studies assign this role to pre-SMA (*Li et al., 2006*; *Nachev et al., 2007*). As an alternative, some authors suggest that the functional role of IFC within response inhibition is either the encoding of context-specific task rules or simply the attentional detection of the need for response inhibition, whereas its initiation is triggered by the pre-SMA (*Duann et al., 2009*; *Sharp et al., 2010*; *Rae et al., 2015*; *Xu et al., 2017*). A more critical view questions the behavioral construct of response inhibition in general (*Hampshire and Sharp, 2015a*). Here, the authors suggest that there is no unitary response inhibition module and claim that the IFC is part of a domain-general control network (*Erika-Florence et al., 2014*). However, in the attempts to experimentally define the differential roles of IFC and pre-SMA, the temporal activation order of those regions has become a decisive, yet unresolved question in the field (*Allen et al., 2018*; *Swann et al., 2012*).

Two methodological issues may explain why this topic has so far remained unresolved. One is a lack of temporal resolution in many respective studies, the other is the specific design of the standard SST. The majority of studies used fMRI, while only few studies in humans addressed the temporal activation order of IFC and pre-SMA using techniques providing both, high spatial and high temporal resolution at the same time. One study used electrocorticography (ECoG) in a single patient and concluded that pre-SMA precedes the IFC activation (*Swann et al., 2012*). In contrast, two recent MEG studies suggest that both regions may be simultaneously active and that there is no temporal or functional primacy (*Allen et al., 2018*; *Jha et al., 2015*). However, the latter two studies may be limited by the design of the standard SST, which does not allow to disentangle attentional from inhibitory processes, and hence, to unequivocally define the functional roles of IFC and pre-SMA. To specifically control for the attentional load involved in the SST, *Sharp et al., 2010* used a selective stopping task. This task comprises additional attentional capture go (AC-GO) trials that are identical to STOP trials in terms of timing and frequency, but participants have to continue with their already prepared go response (*Figure 1A*). Contrasting both conditions (i.e., STOP versus AC-GO trials) has two main advantages. First, it allows us to identify the neural signature of response inhibition independently of other cognitive processes that are related to attention or conflict (*Sharp et al., 2010*). Second, this contrast enables a comparison between conditions where stop and go processes are disturbed in the same way due to a salient stimulus. More importantly, the timing of the initiation of stop and go processes is comparable between conditions. This is not the case when successful stop (sSTOP) trials are contrasted with GO or unsuccessful stop (uSTOP) trials (*Sánchez-Carmona et al., 2016*). Note that the race model assumes that in sSTOP trials, the go process is slower than the stop process, while in uSTOP trials it is vice versa. Since we aimed to analyze the temporal activation order of IFC and pre-SMA with respect to the initiation of response inhibition, sSTOP trials need to be compared to an appropriate control condition that has on average the same signal delay (*Xu et al., 2017*) and is matched in terms of the onset of the related processes. This is particularly important for high temporal resolution studies.

So far, to the best of our knowledge, only three studies used a selective stopping task with an AC-GO ('continue') condition in connection with a high temporal resolution technique. Two studies used ECoG in a single patient (*Swann et al., 2012*), or in four patients (*Wessel et al., 2013*), respectively, but *Wessel et al., 2013* could only cover the right IFC region, not pre-SMA. *Sánchez-Carmona et al., 2016* used EEG, but did not analyze the temporal activation order of both regions. However, the majority of studies providing high temporal resolution relied on a standard SST (*Allen et al., 2018*; *Bartoli et al., 2018*; *Fonken et al., 2016*; *Jha et al., 2015*).

The current study aims to answer long-standing unresolved questions concerning the roles and timing of IFC and pre-SMA during response inhibition, that is, which region initiates inhibitory control and which region exerts a putative causal influence on the other. We therefore used a stimulus selective stopping task in combination with high temporal resolution MEG recordings. To validate

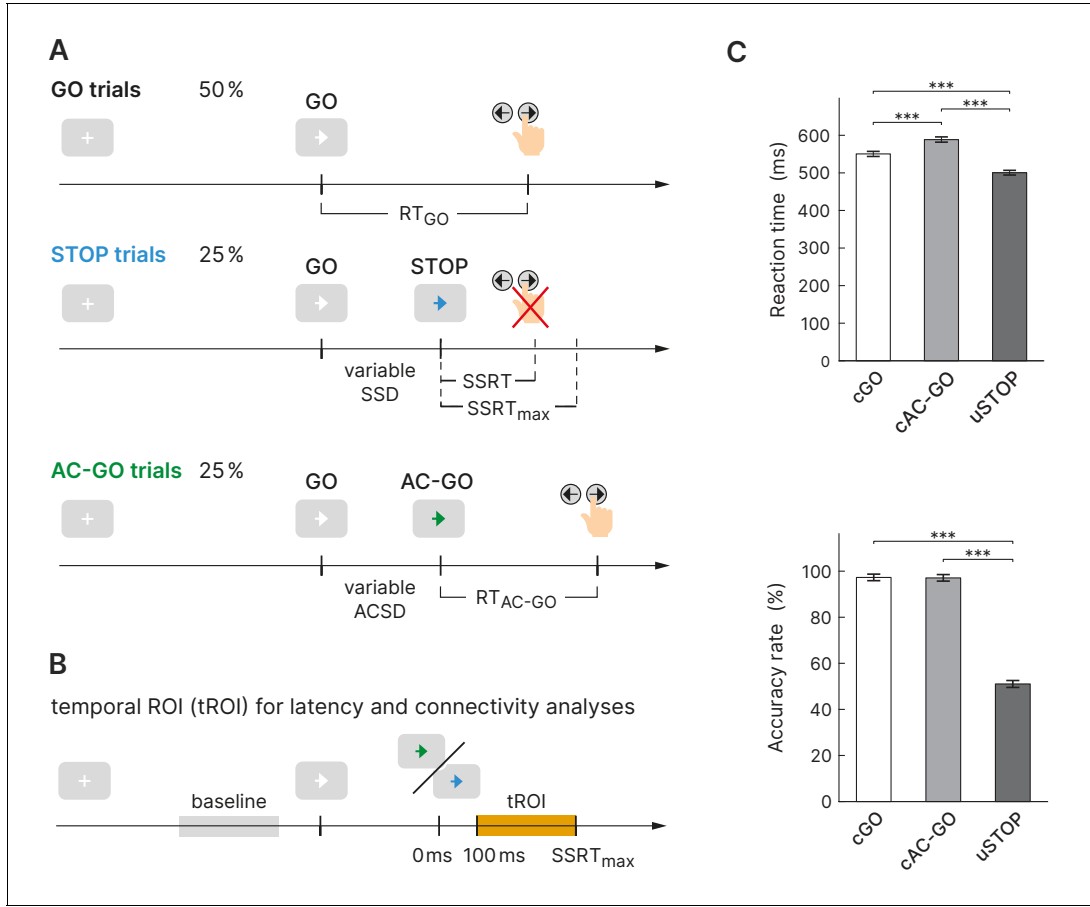

**Figure 1.** Experimental design and behavioral performance of the selective stopping task. (**A**) The task comprised three conditions: a GO condition (white arrow), a STOP condition (arrow changes color to blue), and an attentional capture GO (AC-GO) condition (arrow changes color to green). The variable stop signal delay (SSD) was adapted to the participants' performance to yield a probability of 50% of successful response inhibitions per block. *SSRT*, stop signal reaction time, *ACSD*, attentional capture signal delay (analog to SSD). (**B**) Temporal region of interest (orange box) for latency and connectivity analysis starts at 100 ms, where early visual processing is likely complete and ends at $SSRT_{max}$ (maximal SSRT across participants, 350 ms). (**C**) Reaction time (RT) and accuracy rate in correct GO (cGO), correct attentional capture GO (cAC-GO), and unsuccessful stop trials (uSTOP); mean ± standard error of the mean. ***p<0.001, based on Bonferroni-corrected post hoc comparisons of a repeated-measures ANOVA, *n* = 59.

The online version of this article includes the following source data for figure 1:

**Source data 1.** Behavioral data used for the statistics in panel (C).

MEG source reconstruction, we used the high spatial resolution of fMRI scans in a multimodal imaging approach. Based on previous findings, we hypothesized a predominant role of beta-band activity in both regions during the time of stopping (*Kühn et al., 2004*; *Swann et al., 2009*; *Fonken et al., 2016*) and aimed to define the temporal activation order and connectivity of IFC and pre-SMA during response inhibition.

## Results

### Behavioral results

Healthy participants (*n* = 62; three of them were excluded from further analysis, see 'Materials and methods') performed a stimulus selective stopping task with an attentional control condition (*Figure 1A*). Depending on different signal colors, participants were instructed to respond with a button press, to try to inhibit or to continue their already initiated go response. The mean SSD was 294.5 ± 120.6 ms (mean ± SD) and led to a probability of responding on a STOP trial close to 50% (49.0 ± 3.0%) proving the adherence of participants to the task rules and the successful operation of the staircase procedure (see 'Materials and methods' for more information). While for sSTOP

trials the mean SSD was 288.1 ± 124.6 ms, it was 300.9 ± 117.1 ms for uSTOP trials. The mean attentional capture signal delay (ACSD) for correct attentional capture go (cAC-GO) trials was 296.3 ± 118.4 ms. A repeated-measures ANOVA based on RT revealed a main effect of condition ($F$ = 471.5, p<0.001). Bonferroni-corrected post hoc comparisons revealed that mean RT in cAC-GO trials (588.2 ± 14.0 ms, group mean ± standard error of the mean) was significantly longer as compared to GO trials (550.2 ± 13.5 ms) and uSTOP trials (499.6 ± 12.6 ms). Mean RT in uSTOP trials was significantly shorter as compared to GO trials (p<0.001 for all post hoc tests; *Figure 1C*). Participants performed accurately as indicated by low omission error rates in GO (2.3 ± 2.8%, mean ± SD) and AC-GO trials (1.2 ± 1.4%). Accuracy was 97.2 ± 2.8% in GO and 97.0 ± 2.8% in AC-GO trials (*Figure 1C*). All these behavioral results of the MEG study are in line with the ones of the fMRI study (*Sebastian et al., 2016*). SSRT was calculated using the integration method (see 'Materials and methods' for more information), resulting in a median SSRT of 244.2 ± 36.0 ms. The maximal SSRT across participants (SSRT$_{max}$) was 349.9 ms.

To exclude an effect of signal color, the experiment was designed in a cross-balanced manner, that is, the attribution of signal color for stopping (blue/green) to trial type (STOP/AC-GO) was balanced across subjects. A two-sample *t*-test with signal color as between factor revealed that the stopping latency as measured by the SSRT did not differ significantly between groups ($t$ = 0.088, p = 0.930). A mixed-design ANOVA with signal color as between factor and RT (GO versus AC-GO) as within factor further revealed no influence of attribution of color to trial type, as no interaction of these two factors was present ($F$ = 0.847, p = 0.361).

## Source activation

To analyze the oscillatory dynamics and connectivity of response inhibition, we first identified a set of cortical sources involved in successful stopping. Hence, we contrasted spectral source power of sSTOP and cAC-GO trials. This approach also enabled us to identify coordinates at which stopping-related sources were actually active in the acquired MEG data. This is in contrast to seed-based analyses which rely on a priori defined coordinates from an anatomical atlas (mask) or other studies as reference. Of these two approaches, contrasting appropriate conditions is the recommended best practice (*Gross et al., 2013*). Contrasting conditions enabled us to identify coordinates of peak activity for rIFG and pre-SMA, that is, the locations with the strongest effects across subjects; relying on this approach, compared to a seed-based one, also excluded that the effects we analyzed were due to leakage from other sources.

To find appropriate parameters for source reconstruction using a beamformer method, the spectral power at sensor level was analyzed for both conditions pooled, sSTOP and cAC-GO trials: We a priori defined a temporal region of interest (tROI) starting after the presumed completion of early visual processing (*Amassian et al., 1989*; *Swann et al., 2009*; *Allen et al., 2018*) and ending with SSRT$_{max}$ (100–350 ms, see 'Materials and methods' for more information). We then compared spectral power between the tROI and an equally long baseline epoch (see *Figure 1B*, *Figure 2—figure supplement 1A*) to identify frequency bands of interest for further analyses. The cluster-based permutation test revealed one cluster with significant power decrease (12.0–31.9 Hz, beta band) and one cluster with significant power increase (63.8–87.7 Hz, gamma band).

To then reveal effects related to response inhibition, we reconstructed source power during the tROI and contrasted it between both conditions (sSTOP and cAC-GO) for each of the two frequency bands identified above. This contrast revealed that beta-band power was increased in sSTOP compared to cAC-GO trials in a network of pre-motor and pre-frontal areas (*Figure 2A*). One main cluster of increased beta-band power was located in the IFC with its maximum in the right inferior frontal gyrus (rIFG), pars opercularis. Increased beta-band power was also present in the pre-SMA, left middle frontal gyrus (lMFG), and bilateral premotor regions (*Table 1*, *Figure 2A*). The same analysis performed for the gamma band did not reveal a significant cluster. To further validate the MEG source reconstruction, a subset of participants in the MEG experiment (*n* = 31) plus 45 additional participants were recorded using fMRI (resulting in *n* = 76). Here, the contrast sSTOP versus cAC-GO revealed broad activation of bilateral prefrontal, sensorimotor, parietal, and occipital areas (*Figure 2B*). In particular, rIFG was activated in two subregions (pars triangularis, BA45, x = 52, y = 22, z = 4, t = 10.96; pars opercularis, BA44, x = 52, y = 18, z = 14, t = 9.42), while pre-SMA (x = 16, y = 14, z = 64, t = 9.45) was identified as sub-maxima in the biggest cluster obtained (13,438 voxels) as well as in a smaller cluster (85 voxels) (x = –14, y = 18, z = 66, t = 5.63). For a

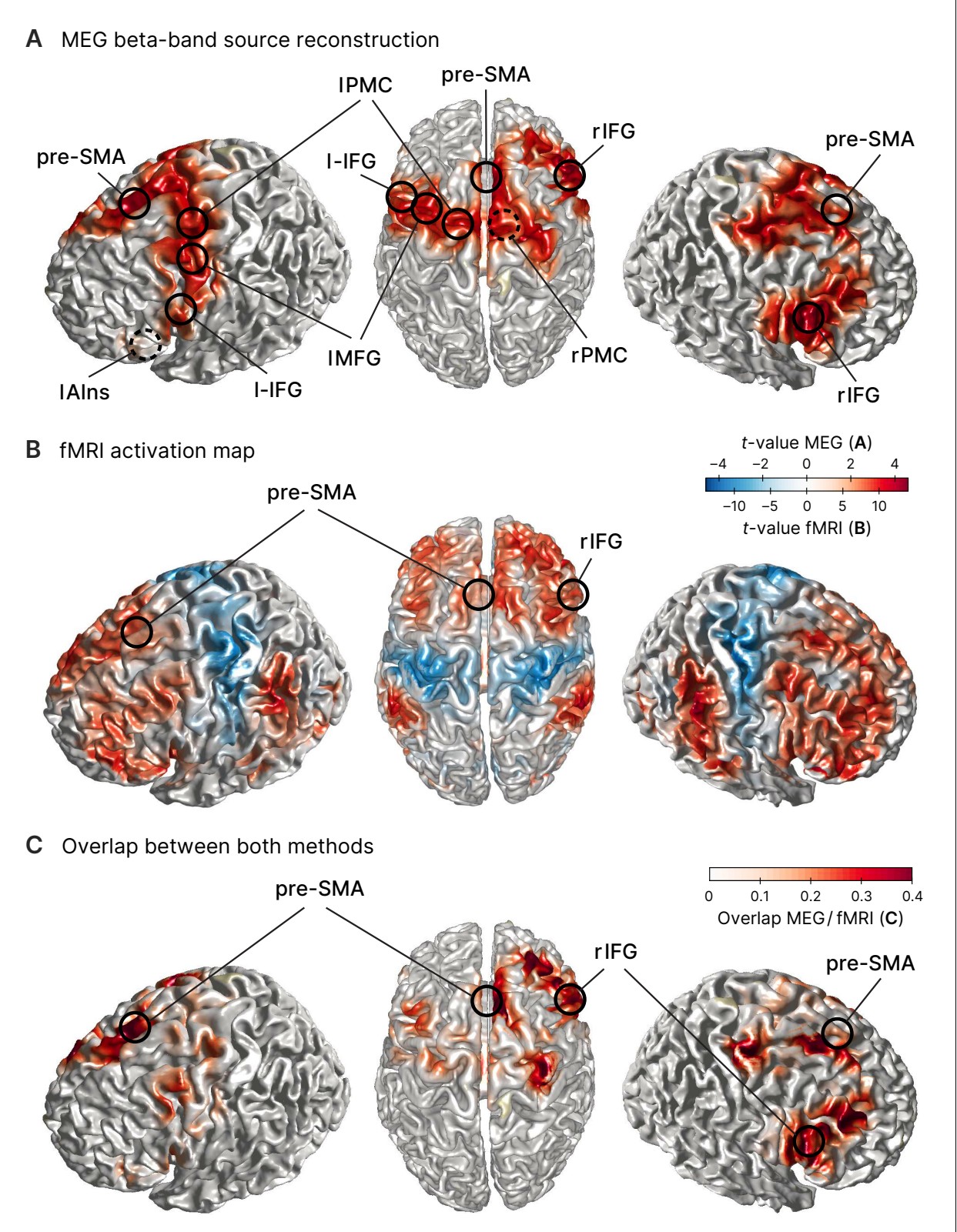

**Figure 2.** Sources involved in successful stopping. (**A**) Source reconstruction of MEG data in the beta band (12–32 Hz, 100–373 ms after STOP/AC-GO [attentional capture go] signal onset). Surface plots show *t*-values of significant clusters when contrasting successful stop (sSTOP) and correct attentional capture go (cAC-GO) trials (cluster-based permutation test, two-tailed, $\alpha_{cluster}$ = 0.05, *n* = 59). Peak voxels (local extrema) of these clusters are highlighted and labeled (MNI coordinates are shown in *Table 1*). Dashed circles indicate that the peak voxel is not directly visible on the surface. *Figure 2 continued on next page*

*Figure 2 continued*

Anatomical regions: *rIFG*, right inferior frontal gyrus, *pre-SMA*, pre-supplementary motor area, *l-IFG*, left inferior frontal gyrus, *lAIns*, left anterior insula, *lMFG*, left middle frontal gyrus, *lPMC*, left premotor cortex, *rPMC*, right premotor cortex. (B) fMRI activation maps for the contrast sSTOP versus cAC-GO; map thresholded $p_{FWE} < 0.05$, cluster extent $k = 5$ voxels, $n = 76$. Positive activation, that is, positive *t*-values (red color), is obtained by the contrast sSTOP > cAC-GO, while negative activation, that is, negative *t*-values (blue color), is obtained by the contrast sSTOP < cAC-GO. Data are taken from *Sebastian et al., 2017*. (C) Overlap between MEG (A) and fMRI (B) activity. To compare the activity revealed by both methods, the product of normalized *t*-values from both methods is displayed. Peak voxels of right inferior frontal gyrus (rIFG) and pre-supplementary motor area (pre-SMA) as found in the MEG analysis are highlighted and labeled.

The online version of this article includes the following figure supplement(s) for figure 2:

**Figure supplement 1.** Activation versus baseline statistics at sensor level.

whole-brain comparison between both methods, MEG and fMRI, we generated an overlap visualization (*Figure 2C*). The overlap was defined by the product of absolute, normalized *t*-values for each voxel $i$, $|t_{i,\mathrm{MEG}}| \times |t_{i,\mathrm{fMRI}}|$. MEG data were interpolated to match the fMRI resolution. Normalization was performed for each method separately by division with the maximum *t*-value obtained by the respective method, resulting in normalized *t*-values between 0 and 1. The comparison of activity revealed by the MEG and fMRI contrasts showed an overall good correspondence. Given the 1 cm grid used for the MEG source reconstruction, the distance between peaks of both methods were rather close for rIFG (7 mm), pre-SMA (15 mm), lAIns (11 mm), and lMFG (15 mm). The peaks of premotor areas (rPMC, lPMC) as well as l-IFG were less close to the fMRI results (20–24 mm difference between methods), but their activity was rather widespread. Yet for the key areas that were most important for this study, rIFG and pre-SMA, we found a very good correspondence and thus consider the MEG source reconstruction as quite accurate for these areas.

## Temporal precedence of rIFG over pre-SMA activation

To identify the cortical region that initiates response inhibition, we analyzed the temporal development of beta-band power across the sources obtained by contrasting *z*-transformed sSTOP with cAC-GO trials. The earliest beta-band power difference was found in the rIFG source compared to all other sources (*Figure 3C*). According to our hypothesis, we then tested explicitly whether the beta-band power difference occurred earlier or later at the rIFG than at the pre-SMA. Since cluster-based permutation tests on time-frequency representations (TFRs) do not establish significance of latency differences between conditions (*Sassenhagen and Draschkow, 2019*), we tested the temporal lead of rIFG activity onset by contrasting *z*-transformed spectral power time courses in sSTOP trials with those in cAC-GO trials. On the spectral power time courses, onset times were defined by using a threshold defined by certain percentage of the range between zero power and the first local power maximum found in the tROI (see 'Materials and methods' for more information). These onsets were then used as input for a permutation test that revealed a significant latency onset difference

**Table 1.** MEG source reconstruction in the beta band.

MNI coordinates obtained as peak voxels when contrasting successful stop (sSTOP) versus correct attentional capture GO (cAC-GO) trials.

| Region (label) | Coordinates | | | *t*-Value |
|---|---|---|---|---|
| | *x* | *y* | *z* | |
| Inferior frontal gyrus BA44/45 R (rIFG) | 50 | 20 | 20 | 3.654 |
| Inferior frontal gyrus BA44/45 L (l-IFG) | −50 | 10 | 20 | 2.820 |
| Anterior insula/left IFG (lAIns) | −40 | 30 | 0 | 2.894 |
| Pre-supplementary motor area (pre-SMA) | 0 | 20 | 60 | 3.878 |
| Middle frontal gyrus (lMFG) | −40 | 0 | 50 | 3.360 |
| Premotor cortex BA6 R (rPMC) | 10 | −10 | 50 | 4.015 |
| Premotor cortex BA6 L (lPMC) | −20 | −10 | 60 | 3.571 |
| Fornix | 0 | 0 | 10 | 3.740 |

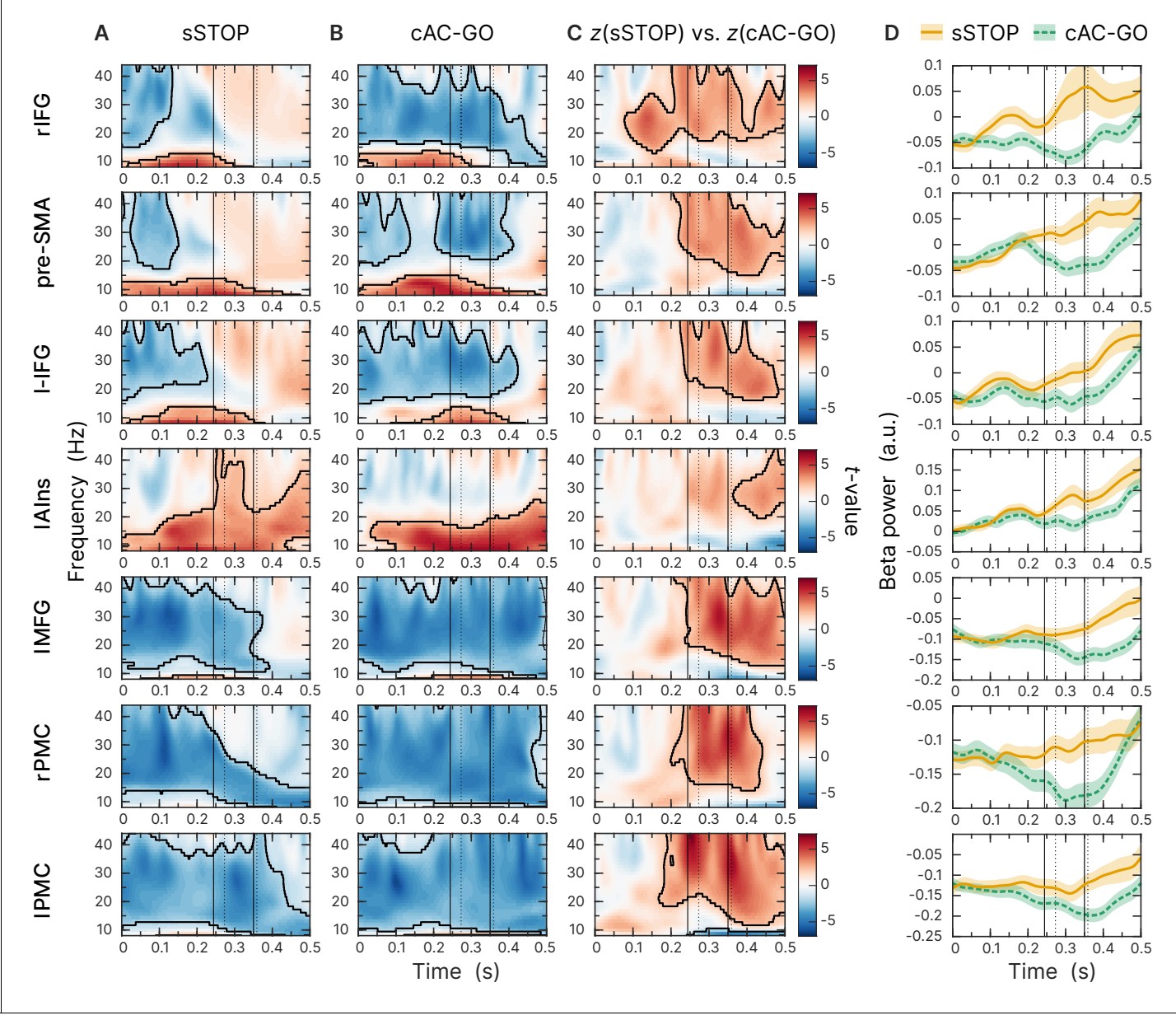

**Figure 3.** Beta-band time-frequency representations of virtual channels placed at sources identified. First and second solid lines indicate stop signal reaction time (SSRT) and SSRT$_{max}$; first and second dotted lines indicate 10% and 50% percentiles of RT$_{AC-GO}$ for selected AC-GO trials with RT$_{AC-GO}$ > SSRT (RT$_{AC-GO}$ is the duration between AC-GO signal and button press, see *Figure 1A*). Time axis locked to STOP and AC-GO signal (0 s). All plots show the results of a cluster-based permutation test, two-tailed, $\alpha_{cluster}$ = 0.05, $n$ = 59, significant clusters are outlined. (A) Task-versus-baseline power for successful stop (sSTOP) trials. (B) Task-versus-baseline power for correct attentional capture go (cAC-GO trials). (C) Contrast of z-transformed sSTOP versus cAC-GO trials. (D) Averaged beta-band power (12–32 Hz). Orange curve (solid line): sSTOP trials, green curve (dashed line): cAC-GO trials, bounded lines: standard error of the mean.

between both sources (p=0.0240, two-tailed test, mean onset latency rIFG: 137 ± 46 ms, and pre-SMA: 159 ± 52 ms). The significant difference reported here was based on a 25% onset threshold, but the onset difference was also significant for 10%, 30%, and 50% thresholds. For this test, seven subjects had to be excluded because no positive peak could be found within the tROI ($n$ = 52).

To further ensure that the earlier onset latency of rIFG compared to pre-SMA does not rely on the specific time-frequency transformation method used and the parameters applied for onset latency definition, we used an orthogonal approach to define the onset latency of both regions. Here, we used broadband signal time courses instead of a frequency-specific analysis. We performed

a time-resolved support vector machine (SVM) analysis in order to classify sSTOP versus cAC-GO trials and to identify the discriminability onset between both conditions, separately for rIFG and pre-SMA. This SVM-based approach revealed a robust above-chance level classification starting from 129 ms (p<0.0001) after STOP/AC-GO signal onset for the rIFG source and from 163 ms (p<0.0001) for the pre-SMA source (*Figure 4A*, tested within tROI, 100–350 ms). To statistically compare individual classification onsets between the two sources, we defined onsets using the final classification time series within the tROI. This analysis showed a significantly earlier rIFG onset compared to pre-SMA at a threshold of 25% with p=0.0377 (p=0.0062). Mean onset in rIFG was 178 ms (186 ms) and 200 ms (220 ms) in pre-SMA (first value refers to an analysis that relied on classification peaks with an accuracy above one standard deviation of the mean baseline accuracy, while the second value in parentheses refers to two standard deviations). Three (17) subjects were excluded because they did not meet inclusion criteria, see 'Materials and methods' for more information. These results were not biased by the selected threshold since all other thresholds gave similar results. Since the onset definition relied on low-pass filtered data for smoothing, onsets in average are later than the values obtained by the above-chance level test. Additionally, to check if this effect was beta-band driven, we performed the same analysis on beta-band filtered data, but still in the time domain. Accuracy was significantly above chance level in the rIFG at 132 ms (p<0.0001) and in pre-SMA at 142 ms (p<0.0001), tested within tROI (*Figure 4B*). However, since the beta-band filtered data was quite noisy, most of the subjects were excluded from the individual onset statistics when applying the same onset definition as for the broadband data. Thus, the onset of rIFG compared to pre-SMA using beta-filtered data was nominally earlier, but this difference did not reach statistical significance. This result indicates that the temporal precedence effect in broadband SVM analysis we found was not only driven by beta-band oscillations. Yet, the beta-band filtered data allowed to reliably classify sSTOP trials, indicating that beta-band oscillations are carrying the crucial information to reliably classify sSTOP trials.

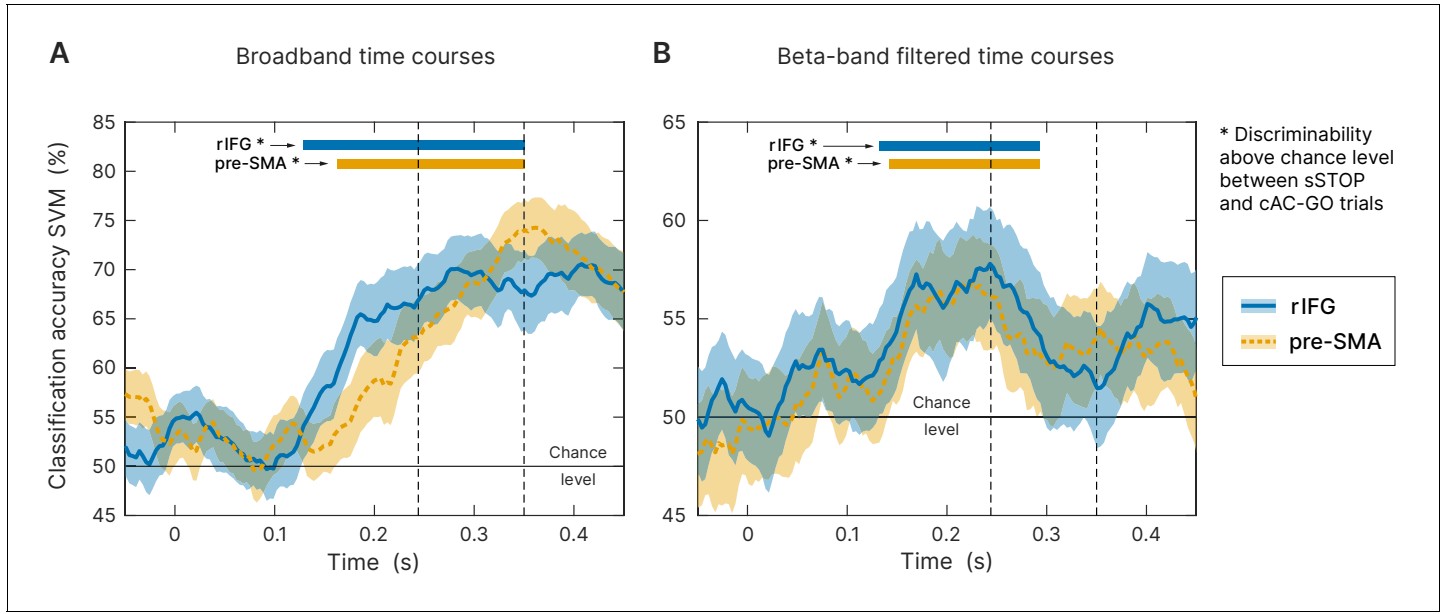

**Figure 4.** Time-resolved support vector machine (SVM) analysis. Both trial types (successful stop [sSTOP] and correct attentional capture go [cAC-GO]) were robustly classified above chance within the temporal region of interest (tROI) (100–350 ms) across subjects at right inferior frontal gyrus (rIFG) (blue curve, solid line) and pre-supplementary motor area (pre-SMA) (orange curve, dashed line). As input, time-embedded vectors of 14 consecutive samples were used, corresponding with one beta cycle at the center frequency of 22 Hz (see 'Materials and methods'). Significance for above chance-level accuracy was tested between 100 and 350 ms. Significant classifications within this range are highlighted by colored bars above the curves and onsets are indicated by arrows. Bounded lines: 95% confidence intervals of the mean, n = 59. First and second dashed lines indicate stop signal reaction time (SSRT) and SSRT$_{max}$. (A) Broadband time course data as input for the SVM. While above-chance level onset for rIFG was at 129 ms (p<0.0001), pre-SMA was above chance at 163 ms (p<0.0001). (B) Same analysis applied on beta-band filtered data (12–32 Hz). While above-chance level onset for rIFG was at 132 ms (p<0.0001), pre-SMA was above chance at 142 ms (p<0.0001).

In sum, the obtained results were comparable between all three analyses, that is, broadband and beta-filtered SVM as well as the onsets obtained by the TFR analysis with respect to an earlier onset in rIFG. Overall, the analysis of time domain data corroborates the significantly earlier activation of rIFG in the beta-band power analysis, suggesting that this area plays a leading role in initiating response inhibition.

## Beta-band source power predicts inhibitory performance

To understand the effects of changes in beta power on the SSRT in more detail, we performed a Bayesian multiple regression analysis with per subject SSRT values as outcome variable and beta power (STOP–cAC-GO) in pre-SMA and rIFG in successful and failed trials, as well as the directed influence asymmetry index (DAI) (see below) as regressors. We compared three models for their expected posterior log-probability using a leave-one-out cross-validation (LOO-CV) procedure as suggested in *Gelman et al., 2014*. Model 1 used all regressor variables and their first-order interaction terms, model 2 used all regressor variables, but no interaction terms, and model 3 additionally averaged together the beta power in pre-SMA for successful and failed trials as the values of these regressors were highly correlated ($r > 0.6$). Thus, model complexity was reduced successively from model 1 to model 3.

Model 3 achieved the best LOO-CV score (expected log posterior predictive density $\mathrm{elpd}_{\mathrm{loo}} = -64.65 \pm 5.32$, higher is better, maximum is at zero), but model 2 was within one standard error of model 3 ($\mathrm{elpd}_{\mathrm{loo}} = -65.24 \pm 5.10$). Model 1 had a considerably worse LOO-CV score ($\mathrm{elpd}_{\mathrm{loo}} = -71.67 \pm 4.53$). In the winning model 3, there was as 91% marginal posterior probability for a negative regression coefficient of the beta power in rIFG for successful trials, and a 87% marginal posterior probability for a negative regression coefficient for the DAI, but only a 64% marginal posterior probability for a negative regression coefficient for the average beta power in pre-SMA, and a 40% marginal posterior probability for a negative regression coefficient for the beta power in rIFG in failed trials (for more details, consult the linked notebook in *Box 1*). Given these results, we consider it most plausible that beta power in rIFG in successful trials and the DAI predict SSRT outcomes, with all other regressors having no or a much smaller influence. This finding strongly supports our hypothesis that modulation of beta-band activity is related to response inhibition.

## rIFG is the dominant sender of information during response inhibition

To further characterize the functional role of rIFG and pre-SMA during response inhibition, we analyzed their connectivity pattern using nonparametric conditional Granger causality (cGC). We contrasted spectral cGC of sSTOP and cAC-GO trials in the tROI. Therefore, we computed the cGC from rIFG to pre-SMA and vice versa, conditional on all other five active cortical sources as obtained by the source reconstruction contrast. The cluster-based permutation test (see 'Materials and methods' for more information) revealed statistical differences between sSTOP and cAC-GO conditions within the (high) beta band in the direction rIFG to pre-SMA ($p_{\mathrm{cluster}} = 0.0074$, corrected for the two links tested), but not in the opposite direction (*Figure 5A*).

To assess if the connectivity between rIFG and pre-SMA was correlated with SSRT, we used the DAI (*Bastos et al., 2015*) that captures the predominant direction of GC and allows comparison of DAIs across frequencies (see Materials and methods). We correlated the DAI of the sSTOP condition with SSRT in the significant frequency range identified (in steps of 2 Hz, in the range from 30 to 40 Hz, resulting in six discrete frequencies; *Figure 5A*). Using a bootstrap method (*Efron and Tibshirani, 1993*), we found a significant negative correlation between DAI and SSRT at 34 Hz ($r = -0.332$, within a 99.17% confidence interval, equivalent to Bonferroni correction for testing six frequencies, [–0.558; –0.053]). Hence, the better subjects were able to inhibit their response, that is, the faster their SSRT was, the higher, positive DAI values they had, indicating a stronger connectivity from rIFG to pre-SMA, whereas slower inhibiting subjects showed the reverse pattern (negative DAI, stronger connectivity from pre-SMA to rIFG, *Figure 5B*). Additionally, this finding was supported by our Bayesian analysis, which showed that the DAI regressor indeed provided relevant information on the SSRT outcomes, even when combined with other relevant regressors in the full Bayesian model. This demonstrates that the asymmetry of communication between rIFG and pre-SMA affects the stopping process.

## Box 1. Toolboxes and custom functions used for analysis of the MEG and fMRI data.

**Toolboxes**
MEG preprocessing, source reconstruction, and TFRs
FieldTrip (*Oostenveld et al., 2011*), FieldTrip: RRID:SCR_004849
https://www.fieldtriptoolbox.org
fMRI preprocessing and activity map
SPM8 toolbox, SPM8: RRID:SCR_007037
http://www.fil.ion.ucl.ac.uk/spm/
Statistics of behavioral data
JASP version 0.12.2, JASP: RRID:SCR_007037 PMID:28685272 (*Allen et al., 2018*)
https://jasp-stats.org/
Support vector machine
Multivariate pattern analysis for MEG (*Guggenmos et al., 2018*)
https://github.com/m-guggenmos/megmvpa , SVM / Megmvpa: no RRID found
LIBSVM: A Library for Support Vector Machines (*Chang and Lin, 2011*)
https://www.csie.ntu.edu.tw/~cjlin/libsvm/, LIBSVM: RRID:SCR_007037
Conditional Granger causality
As implemented in FieldTrip, using a blockwise approach (*Wang et al., 2007*)
https://www.fieldtriptoolbox.org/example/connectivity_conditional_granger/
Bayesian multiple regression
PyMC3 (*Salvatier et al., 2016*)
https://docs.pymc.io, PyMC3: RRID:SCR_018547
**Custom functions**
Sensor statistics
FFTAnalysis.m, FFTStatsTaskvsBaselinePooled.m
Source statistics
Statistics_Step1_Analytic_BC.m, Statistics_Step2_DepSamplesT_BC.m
SourceLocalization.m
TFRs and onset latency analysis
TFRStatsConditionContrast.m, TestOnsetLatencies.m
fMRI activity map
ttest_stopVSac_SPM_job.m
Support vector machine
run_mvpa.m
Conditional Granger causality
masterGC.m
Bayesian multiple regression
MultiRegressionSSRT.ipynb

**Source code** is stored in a GitHub repository:
https://github.com/meglab/acSST (copy archived at swh:1:rev:ea0bf4acc0f11cd-c78ad31b6c1285f1851389312; *Schaum, 2021*)
**Source data** (MATLAB files) are stored at the Dryad open-access repository:
https://doi.org/10.5061/dryad.x3ffbg7gp
For more information, please consult the readme.txt file in the GitHub repository.

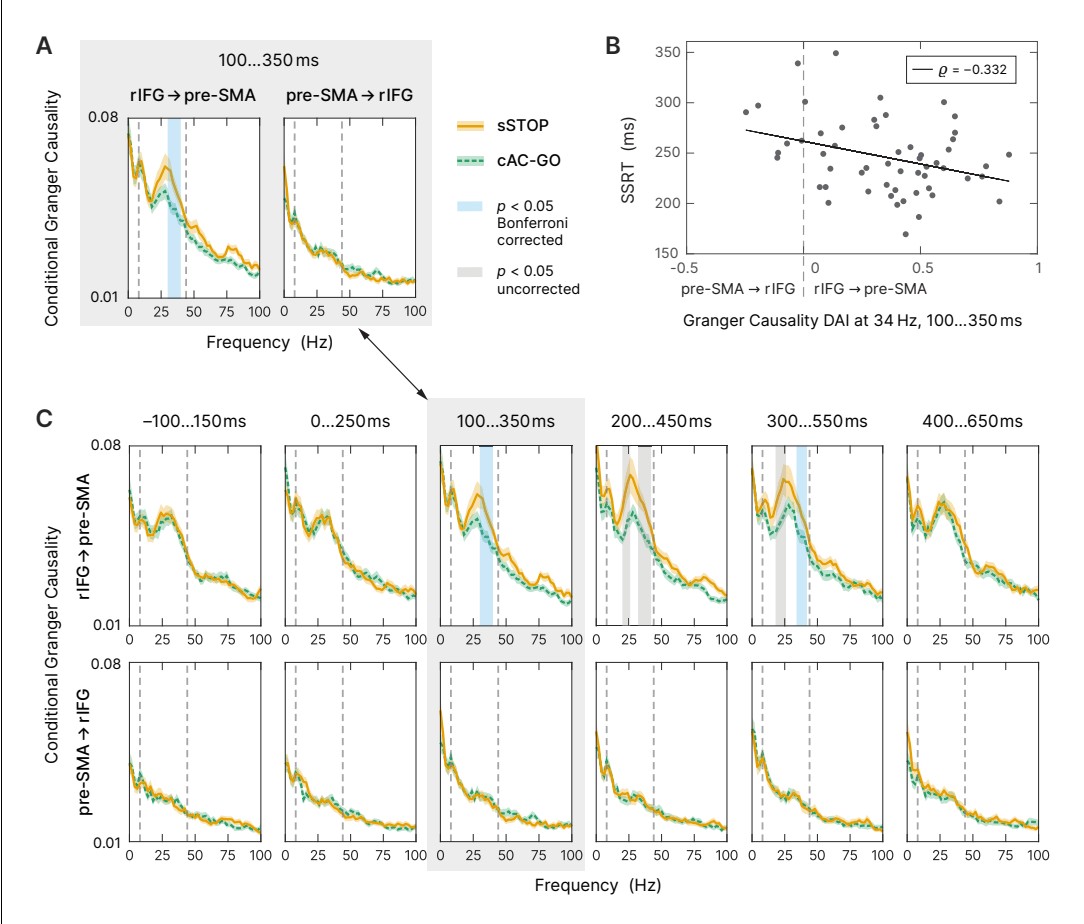

**Figure 5.** Connectivity analysis. (**A**) Spectral-resolved conditional Granger causality (cGC) between right inferior frontal gyrus (rIFG) and pre-supplementary motor area (pre-SMA). Orange curve (solid line): cGC for successful stop (sSTOP) trials, green curve (dashed line): cGC for correct attentional capture go (cAC-GO) trials, bounded lines: standard error of the mean. A cluster-based permutation test was used to identify significant differences between both conditions. Significant differences (p<0.05) are highlighted in blue if Bonferroni corrected, that is, the two links tested, and in gray if uncorrected. The frequency range tested (8–44 Hz) is indicated by vertical dashed lines. $n = 58$ for the link rIFG to pre-SMA, $n = 57$ for pre-SMA to rIFG (subjects with average cGC below bias level in both conditions were excluded). (**B**) Stop-signal reaction time (SSRT) correlated with directed influence asymmetry index (DAI) for rIFG and pre-SMA cGC values (sSTOP trials, cGC at 34 Hz in the temporal region of interest [tROI], 100–350 ms). Positive DAI corresponds to links from rIFG to pre-SMA, while negative DAI corresponds to links from pre-SMA to rIFG. The negative correlation indicates that participants with higher cGC from rIFG to pre-SMA are the better inhibitors (faster SSRT). $n = 55$ with average cGC reliably above bias level (note that only sSTOP trials has been used here). (**C**) Temporal evolution of spectral-resolved cGC was analyzed post hoc between rIFG to pre-SMA (top) and pre-SMA to rIFG (bottom). A sliding window was shifted with 100 ms steps around the tROI. cGC from rIFG to pre-SMA (but not in the opposite direction) was significantly higher for sSTOP compared to cAC-GO trials in the tROI itself and in a later time window tested. Here, we corrected for testing six time windows and for testing in two directions (rIFG to pre-SMA and vice versa). Same formatting as in (**A**). Spectra of the tROI, highlighted with a gray-colored box, are also shown in (**A**).

Finally, to test whether rIFG to pre-SMA connectivity was specific to the tROI of response inhibition, we tested the temporal evolution of the cGC contrast between sSTOP and cAC-GO trials. Therefore, we performed a sliding window analysis, moving backward and forward from the main tROI (100–350 ms) in steps of 100 ms (*Figure 5C*). This post hoc analysis revealed decrease in rIFG to pre-SMA connectivity in two additional windows. In the first additional time window from 200 to 450 ms, the flanks of the beta band showed a difference in cGC between conditions. Note that this first effect was not significant after the correction for multiple comparisons of six time windows and of each direction tested (*p*(lower cluster) = 0.0052, *p*(upper cluster) = 0.0207, both uncorrected). However, in a second, later window (300–550 ms), the upper flank of the beta band showed a significant difference between conditions (*p*(upper cluster) = 0.0384, corrected, and *p*(lower cluster) = 0.0064, uncorrected). No significant differences were found in any time window before the

tROI and after 550 ms, nor in the direction of pre-SMA to rIFG. Overall, our cGC findings suggest that response inhibition is initiated within a crucial time range by specific rIFG to pre-SMA connectivity, the strength of which correlates with behavioral performance.

## Discussion

The neural network mechanisms of response inhibition are central to the understanding of the neurobiology of cognitive control and have been the matter of a long-standing debate. Here, we tested specifically which of the two main regions proposed as the cortical initiator of response inhibition, rIFG and pre-SMA, respectively, is activated first and which exerts a putative causal influence on the other.

Our study, using the high temporal resolution of MEG and a relatively large cohort of subjects, is the first to show a significantly earlier activation in the rIFG compared to the pre-SMA. The onset of beta-band activity in the rIFG was found at ~140 ms after the STOP signal, which is in the same range (~120 ms) as revealed by an independent component analysis (ICA) of a recent EEG study (*Jana et al., 2020*). The onset of beta-band activity in the pre-SMA started at least 22 ms later – as independently verified by a latency analysis of spectral power and a time-resolved SVM analysis. Connectivity analysis showed that rIFG sends information in the beta band for successful stopping to pre-SMA but not vice versa. The behavioral significance of beta-band power in rIFG as well as of the connectivity from rIFG to pre-SMA in the beta band was demonstrated by their prediction of stopping performance.

### rIFG initiates response inhibition

So far, only a limited number of previous studies have investigated the temporal activation order of rIFG and pre-SMA in response inhibition at source level with high temporal resolution (MEG or ECoG) by using an SST (*Allen et al., 2018*; *Jha et al., 2015*; *Swann et al., 2012*). Two of these studies (*Allen et al., 2018*; *Jha et al., 2015*) reported that the pre-SMA and rIFG were simultaneously activated in the time window between STOP signal and SSRT. In contrast to these MEG studies, one ECoG study in a single patient concluded that pre-SMA precedes the IFC activation (*Swann et al., 2012*). There may be three reasons for these diverse findings. First, these previous studies rely on a comparably small sample sizes (*n* = 19, *n* = 9, *n* = 1, respectively) compared to the current study (*n* = 59). A second reason is the use of different SST paradigms, that is, non-selective (standard) SSTs (*Allen et al., 2018*; *Jha et al., 2015*) versus a selective stopping task (*Swann et al., 2012* and current study). Contrasting sSTOP trials with uSTOP or GO trials in non-selective SSTs entails several issues related to separating attentional effects from inhibition (see end of this discussion). A third reason is that results may depend on the way onset latencies are defined (see discussion of limitations by *Bartoli et al., 2018*). We therefore validated our main findings that were based on TFRs via an additional approach. In this additional approach, sSTOP and cAC-GO trials were classified using a time-resolved SVM. The sSTOP and cAC-GO trials could be classified 22–34 ms earlier from rIFG activity than from pre-SMA activity in the SVM analysis, independently replicating our result on latency differences in the TFR analysis. Since the temporal onset of discriminability between conditions was found already before the SSRT, this suggests that rIFG activity is indeed pivotal for initiating stopping.

### Beta-band power in rIFG relates to inhibitory performance

We further hypothesized that if activity in the rIFG is a neural correlate for response inhibition, its power should also predict the inhibitory performance at behavioral level, that is, SSRT. Indeed, SSRT was better predicted by the beta-band power of successful trials in rIFG compared to pre-SMA. In the rIFG, the beta-band power of successful trials was negatively correlated with the SSRT. This is in line with several fMRI studies, where a negative correlation between rIFG activation and SSRT was found (*Aron and Poldrack, 2006*; *Aron et al., 2007*; *Rubia et al., 2007*), suggesting that subjects with greater activation in rIFG inhibited more quickly (shorter SSRT). More importantly, this relationship was also revealed by recent EEG/transcranial magnetic stimulation (TMS) studies (*Hannah et al., 2020*; *Sundby et al., 2021*), which found that the timing of beta bursts correlated with SSRT. Hence, the result of our Bayesian multiple regression additionally supports the dominant role of the rIFG for initiating stopping. Notably, both, sSTOP and uSTOP, trials had the same

temporal activation pattern, that is, rIFG before pre-SMA. However, in the sSTOP trials, the rIFG beta-band power predicted the SSRT better than in the uSTOP trials. Hence, the temporal precedence of rIFG over pre-SMA may indeed be relevant for stopping in general, while at the same time, the degree of control by the rIFG is responsible for the success and the performance of stopping.

## Dominance of rIFG during response inhibition supports its role as controller

To further elucidate the role of rIFG in response inhibition, we analyzed the connectivity between rIFG and pre-SMA using spectrally resolved conditional Granger causality analysis (cGCA). This method measures directed information transfer between two regions, that is, it quantifies predictive relationship within the network (*Granger, 1969*; *Geweke, 1982*). If the rIFG indeed has the function of a top-down controller of response inhibition, then it should exert an ongoing causal influence on the pre-SMA across the time window of response inhibition. Indeed, a significant difference between sSTOP and cAC-GO in cGC influence from rIFG to pre-SMA was detected, but not in the opposite direction. In agreement with our hypothesis, significant results were found within the beta band in particular. The observed directionality is in line with cGCA in a comparably high-powered fMRI study (*Duann et al., 2009*) ($n = 57$), yet inconsistent with GCA of a somewhat lower powered MEG study (*Allen et al., 2018*) ($n = 19$). In addition to the finding of significant beta-band cGC from rIFG to pre-SMA, we found that the strength of the directional asymmetry in cGC (DAI) predicted stopping performance as measured by SSRT. This is compatible with a recent DCM (dynamic causal modeling) analysis of an EEG study available in preprint (*Fine et al., 2019*) that also found (inhibitory) connectivity from rIFG to pre-SMA during STOP trials to be negatively related to SSRT when transcranial focused ultrasound was applied. These findings suggest that connectivity directed from rIFG to pre-SMA per se reflects a state in which rIFG is prepared to send a signal to pre-SMA to execute stopping, while the strength of this directionality influences the performance of stopping. To test whether the directionality from rIFG to pre-SMA was specific for response inhibition, or just related to other processes, we analyzed connectivity between rIFG and pre-SMA before and after the (a priori defined) tROI (100–350 ms) using a sliding window approach and indeed found the dominant effects centered on the tROI. Thus, we could confirm that inhibition-related beta-band connectivity between rIFG and pre-SMA was specific to the tROI and a subsequent time window, particularly only in the direction from rIFG to pre-SMA.

In sum, the results of the spectrally resolved cGCA further support the leading role of rIFG in response inhibition and the importance of beta-band oscillations in this process. Since connectivity from rIFG to pre-SMA was specifically established during the tROI and correlated with inhibitory performance, rIFG may be characterized as a brake in response inhibition. This interpretation is in line with *Aron et al., 2014* and was recently corroborated by an ECoG study that revealed evidence for the prefrontal-subthalamic hyperdirect pathway, where stopping-related potentials in the rIFG preceded the ones in the subthalamic nucleus (STN) (*Chen et al., 2020*).

## Involvement of beta-band oscillations in response inhibition

Our results strongly support the interpretation that the rIFG signals the need to stop an already initiated action in the sense that it triggers a cascade of events that ultimately results in action cancelation and that beta-band oscillations are critically involved in this process. In general, beta-band oscillations are thought to be related to the maintenance of the current sensorimotor or cognitive state as hypothesized by a prominent model (*Engel and Fries, 2010*). In contrast, desynchronization of beta-band oscillations enables the transition into an active processing state (*Miller et al., 2012*; *Little et al., 2019*), for example, pressing a button. Our TFR results can be explained within the framework of this model, that is, beta-band desynchronization (BBD) is related to motor response preparation and BBD should already set in early, after the initial GO signal. And indeed, until approximately 100 ms after the STOP/AC-GO signal, we observe almost the same strength of BBD for both conditions (so that it cancels out in the contrast, *Figure 3C*). While in cAC-GO trials, the BBD continues until the median RT in the rIFG (*Figure 3B, D*), BBD is aborted in sSTOP trials already before the median SSRT, as we see an increase of beta-band power (*Figure 3A, D*). As soon as the stopping or motor response, respectively, is completed, beta-band oscillations resynchronize (*Fonken et al., 2016*).

To interpret the role of beta-band oscillations during the initiation of response inhibition, it is important to focus on the early changes in beta-band power, which set in before SSRT and the later so-called beta rebound. With the BBD signaling an ongoing motor response, a slowing or plateauing of BBD in the time period before SSRT could be interpreted as a slowing or pausing of this ongoing motor initiation process. Hence, the reduction of BBD during an ongoing motor initiation process could not be regarded as a distinct active stopping process. Yet, what we found in the rIFG was a rather early termination of BBD in sSTOP trials in the sense that beta-band power started to increase shortly after the STOP signal (*Figure 3D*). However, this increase in synchronization was not strong enough such that beta power was increased above baseline, but values reached baseline levels in rIFG. Thus, the relative change in beta power that we found in the contrast originates genuinely in the sSTOP trials, where BBD is terminated early, that is, before the SSRT, while in the cAC-GO trials no such termination was observed. This finding suggests an active change of processing in rIFG that is related to stopping, while processing proceeds 'as usual' in cAC-GO trials, and the motor response will be executed.

Furthermore, while we are fully aware that absence of evidence is not evidence of absence, the finding of significant Granger causality in time windows relevant for stopping, but not earlier, suggests that active signaling from rIFG to pre-SMA specifically in the beta band is an important part of the neural dynamics underlying stopping. In sum, the two findings, that is, rIFG as source of stopping relevant information transfer in the beta band and a termination of BBD specifically in rIFG already before SSRT, suggest that response inhibition is implemented in part by beta-band modulations. Although the present data and evidence from previous research (*Wessel et al., 2013*; *Castiglione et al., 2019*) indicate that the stopping process is related to a modulation of oscillatory power in the beta band, it still needs to be determined whether beta changes result from a pausing of action initiation or whether they have a genuine stopping function.

## Controlling for attention in a selective stopping task suggests that rIFG does not serve an attentional role

A limitation of most previous neuroimaging studies on response inhibition is that sSTOP trials are usually contrasted with GO or uSTOP trials to identify inhibitory processes. Both contrasts are indeed capable of removing basic perceptual activity due the presentation of the stimuli (*Aron and Poldrack, 2006*; *Sharp et al., 2010*), but leave other confounding processes in place: Contrasting sSTOP with uSTOP trials, for example, involves error processing (*Rubia et al., 2003*) and an early response in motor cortices that has the potential to obscure activity related to the adjacent pre-SMA (*Allen et al., 2018*; *Jerbi et al., 2007*). Additionally, this contrast was criticized as overly conservative, because it assumes that inhibitory processes are also present in uSTOP trials (*Boehler et al., 2010*). Contrasting sSTOP with GO trials instead also introduces two issues: First, a less frequent event like the STOP signal is perceptually more salient and causes significant brain activation that is not directly based on inhibitory processing alone (*Sharp et al., 2010*). Second, STOP signals are delayed compared to the GO signals (i.e., SSD), hence directly contrasting both trial types entails the comparison of different time points within the stopping process (*Figure 1A*). This is particularly important when high temporal resolution methods are used. Additionally, attentional and discriminative processing of the STOP signal are presumed to disturb the timing of go and stop processes compared to GO trials (*Sharp et al., 2010*).

All of the above issues can be resolved by contrasting sSTOP with cAC-GO trials. First, the influence of error processing was reduced to a minimum when the contrast did not rely on unsuccessful trials (*Rubia et al., 2003*). Second, compared to the contrast with GO trials, frequency and stimulus properties of STOP and AC-GO signals could be matched precisely. As stimuli for STOP and AC-GO signals, blue and green arrows were used, respectively, and balanced across subjects. And indeed, the signal color had no effect on SSRT or with respect to RTs, on trial type (GO versus AC-GO). Additionally, trial number was balanced between both conditions, STOP and AC-GO. Hence, both trial types were presented with the same frequency, but still unexpected compared to GO trials. Thus, the stimuli of STOP and AC-GO could be regarded as equally salient. Third, AC-GO stimuli were presented after a signal delay that was adapted using a staircase procedure as for the STOP trials. Thus, timing of cAC-GO trials was matched to the one of sSTOP trials. Additionally, the attentional influence of the signal delay could be confirmed by our behavioral data. We found an increase in RTs after the appearance of an AC-GO signal compared to simple GO trials as predicted. There is

independent evidence that the prolonged time required for the perceptual processing of the AC-GO signal is approximately the same as for the STOP signal compared to GO trials (*Xu et al., 2017*). In this study, the authors demonstrated at least the behavioral similarity of both conditions by comparing SSRT and 'continue-signal reaction time' (i.e., approximately, the time between AC-GO signal and response, a measure adopted from *Mayse et al., 2014*).

Hence, the contrast sSTOP versus cAC-GO is the most appropriate to identify the neural correlates of 'pure' response inhibition, independent of attentional and error processing (*Sánchez-Carmona et al., 2016*). The dissociation of inhibitory from attentional processes by an appropriate task design suggests that the rIFG does not, at least not primarily, serve an attentional role in response inhibition. However, although the rIFG has often been regarded as candidate for an inhibitory module and our results showed that response inhibition is indeed initiated by the rIFG, the rIFG might still be part of a more domain-general cognitive control mechanism, implemented by fronto-parietal networks (*Erika-Florence et al., 2014*; *Hampshire and Sharp, 2015a*).

## Limitations

One limitation of this study is that we could not analyze activity of the STN which is an important subcortical node within the response inhibition network because of its direct relation to IFG (*Aron et al., 2007*; *Duann et al., 2009*; *Jahfari et al., 2011*; *Rae et al., 2015*; *Xu et al., 2017*; *Chen et al., 2020*). No STN activity was detected by our source reconstruction contrast, most likely because MEG is less sensitive to subcortical compared to cortical sources (*Gross, 2019*). Nonetheless, several studies with intracranial recordings detected beta-band oscillations within the STN (*Kühn et al., 2004*; *Ray et al., 2012*; *Zavala et al., 2018*; *Fischer et al., 2018*; *Chen et al., 2020*), also supporting the importance of beta-band oscillations in response inhibition.

## Conclusion

Our findings strongly support the hypothesis that response inhibition is initiated by the rIFG and implemented as a top-down control via beta-band oscillations (*Aron et al., 2014*). This brake is turned on in a time window around 100–200 ms after the STOP signal. Within the framework of the race model, rIFG may promote stopping processes and then signals to pre-SMA, among other regions, to proceed and execute the stopping process. Due to our task design, that is, contrasting sSTOP with cAC-GO trials, we could exclude a primarily attentional role of the rIFG in stopping. Thus, these results may open up new ways of spatially localized oscillation-based interventions, that is, modulating beta-band activity by real-time EEG-triggered TMS. Additional causal evidence for the mechanisms of response inhibition may open up new avenues of novel therapeutic strategies for diseases related to impaired response inhibition.

# Materials and methods

## Participants

Sixty-two healthy subjects participated in the MEG experiment. As a result of applying behavioral outlier criteria (*Congdon et al., 2012*), one participant had to be excluded because the inhibition stop rate was below 40%. A second participant had to be excluded because the mean RT of uSTOP trials was higher than the mean RT of cGO trials, indicating that the assumptions underlying the horse race model (*Logan et al., 1984*) were violated in this participant. A third participant had to be excluded because no head model could be obtained from the available anatomical MR image. All of the remaining 59 participants (39 women; mean age ± SD, 25 ± 6 years) had normal or corrected-to-normal vision. All individual participants included in the study were screened for factors contradicting MRI and MEG scanning and provided written informed consent before participation and consent to publish any research findings based on their provided data in anonymized form. The study was approved by the local ethics committees (Johann Wolfgang Goethe University, Frankfurt, Germany, and Medical Board of Rhineland-Palatinate, Mainz, Germany), and participants were financially compensated for their time.

## Behavioral

SSRT was calculated using the integration method (*Verbruggen et al., 2019*). This method estimates the SSRT by finding the point at which the integral over the RT distribution equals the actual p(respond|signal) and then subtracting the mean SSD. The integration approximately corresponds to the *n*th RT of the distribution of GO trials (sorted by RT), multiplied by the actual p(respond|signal). For instance, p(respond|signal) is 0.48 across 1000 GO trials acquired, then the *n*th RT is the 480th fastest GO RT. Then SSRT is calculated by subtracting the mean SSD from the 480th RT. The distribution of GO trials also includes incorrect GO trials; GO omissions were replaced with the maximum RT in order to compensate for the lacking response (*Verbruggen et al., 2019*). Behavioral data was extracted from the MEG triggers and analyzed with the JASP software (version 0.12.2, https://jasp-stats.org).

## Tasks and procedure

Participants performed a stimulus selective stopping task with an additional attentional control condition, that is, AC-GO trials. The task and procedure was the same as described in *Sebastian et al., 2016*. For stimulus presentation, we used Presentation software (version 13.1, http://www.neurobs.com).

Before the acquisition session, participants had to read the instructions and practice the task on a laptop computer for approximately 5 min. The acquisition session was split into 10 blocks. Throughout the acquisition, participants were asked to respond to the stimuli by pressing a response button with the left or right thumb (MEG) or index finger (fMRI), respectively.

The task comprised three conditions: a GO condition (50% of all trials), a STOP condition (25% of all trials), and an AC-GO condition (25% of all trials). Trials of all conditions began with the presentation of a white fixation cross in the center of the screen with a randomly varied duration between 1500 and 2000 ms in the MEG experiment and 500 ms in the fMRI experiment, respectively (*Sebastian et al., 2016*). Then, a white arrow (GO signal) pointing to the left or right was displayed. Participants were instructed to respond with a left button press to a left pointing arrow and with a right button press for a right pointing arrow. In the GO condition, the white arrow was displayed for 1000 ms (equivalent to the maximum permitted RT) or until a button press was performed. In the STOP condition, the white arrow was presented first, followed by a change of its color from white to blue after a variable SSD. Participants were instructed to try to inhibit their initiated button response after the GO signal. The SSD was adapted to the participants' performance to yield a probability of 50% of successful inhibitions per block. Therefore, a staircase procedure was implemented with the following properties: The initial SSD was set to 210 ms. If the response was not successfully inhibited (unsuccessful stop trial, uSTOP), the SSD in the next STOP trial was decreased by 30 ms with a minimum SSD of 40 ms. If a response was successfully inhibited (successful stop trial, sSTOP), the SSD in the next STOP trial was increased by 30 ms. The maximum SSD was limited by the maximum permitted RT of 1000 ms. In the AC-GO condition, the white arrow was presented first, followed by a change of its color from white to green after a variable AC-GO signal delay (ACSD), and participants were instructed to continue their response. The ACSD was varied in accordance with the staircase in the STOP condition.

The attribution of color (green/blue) to trial type (STOP/AC-GO) was counterbalanced across participants. In case of an omission error (no button press) in the GO or AC-GO condition, participants were given a short feedback ('oops' – no button press') to maintain the participants' attention and to limit proactive slowing. The length of the intertrial interval (presentation of a blank screen) was 700 ms in the MEG experiment. In the fMRI experiment, the length of the intertrial interval was varied randomly between 2500 and 3500 ms. One block consisted of 112 trials presented in a randomized order.

## fMRI experiment

A subset of 31 participants of the MEG experiment performed the fMRI task and additionally data from 45 participants from the *Sebastian et al., 2017* study were included for analysis (*n* = 76). For all participants of the MEG experiment (*n* = 62), an anatomical MRI was acquired on a Magnetom Trio Syngo 3 T system (Siemens Medical Solutions) at two sites (8-channel head coil at site 1 and a 32-channel head coil at site 2). For recording details and analyses, refer to *Sebastian et al., 2017*.

## MEG experiment

### MEG data acquisition and preprocessing

MEG data acquisition was conducted in line with the good practice guidelines for MEG recordings recommended by *Gross et al., 2013*. MEG signals were recorded using a whole-head system (Omega 2005; VSM MedTech) with 275 channels. The signals were recorded at a sampling rate of 1200 Hz in a third-order synthetic gradiometer configuration and were filtered with fourth-order Butterworth 300 Hz low-pass and 0.1 Hz high-pass filters. In addition to the MEG, we recorded the electrooculogram (EOG) and electrocardigram (ECG) for offline artifact rejection. All analyses were performed in MATLAB (MathWorks Inc, Natick, MA), mainly using the FieldTrip toolbox (*Oostenveld et al., 2011*) and custom scripts (see *Box 1*).

Trial definition and selective filtering was directed by our aim to identify neural networks related to response inhibition. To eliminate attentional processes, we intended to directly contrast sSTOP with cAC-GO trials. However, we had to take into account that cAC-GO trials might in turn involve button press-related motor activity that is not present in the sSTOP trials. Therefore, the definition of a tROI should preferably end before button presses in cAC-GO trials set in. However, regarding sSTOP trials, the tROI should not end before response inhibition is initiated and finished. To also account for participants with longer SSRTs compared to the average SSRT, we additionally sought to ensure that for these participants the inhibition response process is still captured by the tROI, so we a priori defined a tROI from 100 ms to an $SSRT_{max}$ of 350 ms relative to the STOP signal. To reduce the contamination of our tROI with button press-related motor activity, we rejected cAC-GO trials with an $RT_{AC-GO} < SSRT$, where $RT_{AC-GO}$ is the 'attentional capture GO RT', defined as duration between AC-GO signal and button press, and SSRT is the median SSRT across subjects (244 ms). More importantly, when contrasting sSTOP and cAC-GO trials, these trials should also be matched in terms of the underlying distribution of go and stop processes according to the independent horse race model. This is important because the neural correlates of response inhibition revealed by this contrast should not rely on mere timing differences between the underlying go and stop processes (*Sánchez-Carmona et al., 2016*). Although the distribution of go and stop processes cannot be directly accessed, the horse race model implies that in sSTOP trials, the response can be successfully inhibited because the go process is slower than the stop process. Thus, to reduce processing speed differences to a minimum, it is necessary to select cAC-GO trials with longer than average RTs, where the underlying distribution consists of slower go processes, comparable with sSTOP trials. Thus, rejecting cAC-GO trials with an $RT_{AC-GO} < SSRT$ additionally supports a close match between the underlying distribution of go and stop processes in both trial types. After this trial selection was performed, the 10% percentile of $RT_{AC-GO}$ was 273 ms, and the median was 359 ms (as indicated in *Figure 3*). To exclude that these results were biased by the cAC-GO trial selection (i. e., only including trials with $RT_{AC-GO} > SSRT$), we performed all analyses on randomly selected cAC-GO trials. Overall, these results were comparable to the reported ones.

Trials containing sensor jump or muscle artifacts were rejected using automatic artifact detection functions provided by FieldTrip. Power line noise was reduced using a discrete Fourier transform filter at 50, 100, and 150 Hz. In addition, to remove artifacts related to eye movements and heart beat, ICA was performed on all trial types (*Makeig et al., 1995*). Data was downsampled to 400 Hz before the extended infomax (runica/binica) algorithm was applied to decompose the data (as provided by FieldTrip/EEGLAB). Independent components (ICs) were removed from the data if their spatial topography corresponded with the artifact type (*Fatima et al., 2013*), which was visually inspected and, when in doubt, correlation coefficients with EOG and ECG data were consulted. Typically, five ICs were rejected (range 2–8 ICs). Since minimization of head movement is crucial to MEG data quality (*Gross et al., 2013*), trials were rejected when the head position deviated more than 5 mm from the mean head position over all blocks for each participant.

### Spectral analysis at sensor level

The only statistics performed at the sensor level was to define appropriate frequency bands as beamformer parameters for subsequent source reconstruction. We therefore compared the spectral power between 4 and 120 Hz (based on Hanning tapers) over averaged channels during the tROI (100–350 ms) with the spectral power of corresponding baseline segments for all participants. For this purpose we used a dependent-samples permutation *t*-test and a cluster-based correction

method (*Maris and Oostenveld, 2007*) to account for multiple comparisons across frequencies. Samples whose $t$-values exceeded a threshold of $\alpha_{\mathrm{cluster}} = 0.01$ were considered as candidate members of clusters of spectrally adjacent samples. The sum of $t$-values within every cluster, that is, the 'cluster size', was calculated as test statistics. These cluster sizes were then tested (two-sided) against the distribution of cluster sizes obtained for 5000 partitions with randomly assigned task and baseline data within each subject. Absolute cluster values above the 99.5th percentiles of the distribution of cluster sizes obtained for the permuted datasets were considered significant. This corresponds with an alpha threshold of $\alpha = 0.01$ that was Bonferroni-corrected for a two-tailed test. Since we aimed to contrast neural power of sSTOP and cAC-GO trials on source level to obtain a set of sources related to response inhibition, we had to use an (orthogonal) statistical test at sensor level in order to avoid circularity (*Kriegeskorte et al., 2009*). We therefore combined both conditions (sSTOP and cAC-GO trials) for analysis on the sensor level and contrasted these 'pooled' task segments with corresponding baseline segments of both conditions.

## Source imaging

The anatomical MRI of each participant was linearly transformed to a segmented standard T1 template of the SPM8 toolbox (http://www.fil.ion.ucl.ac.uk/spm/) in MNI space (*Collins et al., 1994*). This template was overlaid with a regular dipole grid (spacing 1 cm). The inverse of the obtained linear transformation was then applied to this dipole grid and the lead field matrix was computed for each of the grid points of the warped grid using a single shell volume conductor model (*Nolte, 2003*). Since all grid locations of each subject were aligned to the same anatomical brain compartments of the template, corresponding brain locations could be statistically compared over all subjects.

To reconstruct sources in the beta and gamma band, a frequency domain beamformer source analysis was performed by using the dynamic imaging of coherent sources algorithm (*Gross et al., 2001*) implemented in the FieldTrip toolbox. Since our experimental setup did not contain any (coherent) external reference, filter coefficients were constrained to be real-valued to restrict our analysis to local source power. This can be done, because we assume that the magnetic fields propagate instantaneously from source to sensor, and therefore, no phase shifts can occur that would lead to complex coefficients (*Nunez and Srinivasan, 2006*). Beamformer analysis uses an adaptive spatial filter to estimate the power at every specific (grid) location of the brain. The spatial filter was constructed from the individual lead fields and the cross-spectral density (CSD) matrix for each subject. CSD matrices were computed for the task period of 100–373 ms after STOP/AC-GO signal onset and a baseline period of same length, with an offset of −100 ms relative to GO signal onset (see *Figure 1B* for timescale). To avoid spectral leakage, the length of the time-frequency window was extended from 350 ms so that it matches an integer number of oscillatory cycles of the center frequency of the frequency band width (*Harris, 1978*). Frequency bands were defined based on the statistical analysis of spectral power at the sensor level. Thus, CSD matrices were computed in the beta band for 22 Hz (±10 Hz) and the gamma band for 76 Hz (±12 Hz), where spectral smoothing is indicated in brackets. CSD matrix calculation was performed using the FieldTrip toolbox with the multitaper method (*Percival and Walden, 1993*) using four Slepian tapers (*Slepian, 1978*) for the beta band and five Slepian tapers for the gamma band.

## Source statistics

To reveal sources related to response inhibition at group level, we statistically tested differences between source power of sSTOP and cAC-GO trials. First, we calculated an activation-versus-baseline $t$-statistic at single-subject level by using an analytic dependent-samples within-trial $t$-test. The $t$-values obtained from this activation-versus-baseline test were then used as input for a cluster-based permutation test for contrasting ('baseline corrected') sSTOP and cAC-GO trials (dependent-samples $t$-test, Monte Carlo estimate, same two-sided test as described for the spectral analysis at sensor level, but for clusters of adjacent voxels as samples and $\alpha = 0.05$). In the clusters obtained, we searched for local extrema to identify peak voxels. All of the peak voxels obtained were reported, but for subsequent TFR analysis, only physiologically plausible voxels, that is, voxels in gray matter according to the Jülich Histological atlas provided by the FSL toolbox (FMRIB's Software Library) (*Smith et al., 2004*), were used.

## Latency analysis
### TFRs on virtual channel time courses
To analyze the temporal activation pattern of each peak voxel identified by contrasting sSTOP and cAC-GO trials, we computed TFRs of these sources. Therefore, time courses of these sources were reconstructed as 'virtual channels' using bandpass-filtered raw data (2–150 Hz). For time course reconstruction, a time-domain beamformer was used (linear constrained minimum variance) (*Van Veen et al., 1997*), based on common filters analog to the ones used in the frequency domain. In contrast to the frequency domain, uSTOP trials were additionally included in the common filter computation when a post hoc analysis with uSTOP trials was performed. A principal component analysis on the three reconstructed time courses in $x$, $y$, and $z$ direction was performed for each grid point in order to determine the dominant dipole orientation (direction with the largest variance). The time course of the first principal component was used for subsequent TFR analysis (8–44 Hz, Hanning taper with a sliding time window of three cycles were used). We computed three different TFRs for each source: (1) task-versus-baseline power for sSTOP trials, (2) task-versus-baseline power for cAC-GO trials, and (3) the contrast of sSTOP versus cAC-GO trials to reveal significant inhibition-related power changes at group level. In case of the latter TFR, for baseline correction, we $z$-transformed sSTOP and cAC-GO trials of each participant separately by subtracting the mean of the corresponding baseline and dividing by the standard deviation of combined (sSTOP and cAC-GO) baseline trials. For all of the three TFRs, we used a cluster-based permutation test as described for the spectral analysis at sensor level, except for (time,frequency)-pairs serving as samples and $\alpha = 0.05$. However, since cluster-based permutation tests on TFRs do not establish significance of latency differences between conditions (*Sassenhagen and Draschkow, 2019*), we used individually defined onset values to test for a significant latency difference between rIFG and pre-SMA. These individual onset latency (and power) values were defined by subtracting averaged $z$-transformed cAC-GO trials from averaged $z$-transformed sSTOP trials (see below).

### TFR-based latency analysis
To answer the question of the temporal order in which the two candidate sources, rIFG and pre-SMA, are activated, we performed an onset latency analysis based on difference TFRs ($z$(sSTOP) – $z$(cAC-GO)). Source power was averaged over the beta band (12–32 Hz), smoothed and analyzed for peaks within the tROI. Onset latency (for each participant and source) was defined as the time point at which the averaged source power exceeded a threshold of 25% of the range between the first positive peak within the tROI and the (positive) minimum found until this peak (in most cases, zero). If there was no positive peak within the tROI for at least one source, we excluded this subject for the TFR-based latency analysis. To test for differences between onset latencies of rIFG and pre-SMA at group level, a permutation statistics was performed (50,000 permutations). Onset latency differences were also tested with thresholds of 10%, 30%, 50%, 75%, and 100%.

### Alternative latency analysis using time-resolved SVM
To determine the onset of discriminability in the time-domain between sSTOP and cAC-GO trials, we used a time-resolved multivariate support vector machine analysis (t-SVM) with a linear kernel. The t-SVM was trained and tested separately for each subject, and for rIFG and pre-SMA. First, we $z$-normalized each source signal relative to baseline period (samples before the GO cue, 500 ms length). Next, to reduce the computational cost and increase the signal-to-noise ratio, the data were smoothed with a Gaussian Kernel of ±10 ms and downsampled to 300 Hz, similar to the approach in *Hebart et al., 2018*. For time embedding, we used 14 consecutive samples, corresponding to one beta cycle (center frequency 22 Hz). After preprocessing, the embedded source data were randomly assigned to one of eight *supertrials* (per condition) and averaged (MATLAB code was adopted from *Guggenmos et al., 2018*, which builds on the library for SVMs by *Chang and Lin, 2011*). Finally, we separated these *supertrials* into training and testing data with one *supertrial* per condition serving as test data and all the other as training data. The binary classification was performed for each time point from −100 ms before to 700 ms after signal onset. To obtain a more robust estimate of the classification accuracy, we performed 100 iterations of *supertrials* averaging and classification. The final classification time series reflects the average across these iterations. For statistical testing, we employed a nonparametric cluster permutation approach (*Maris and Oostenveld, 2007*), with

clustering of subsequent time points (10,000 permutations). The null hypothesis of no experimental effect for the classification time series was equal to 50% chance level and tested within the tROI (100–350 ms, where 350 ms corresponds to $SSRT_{max}$). Using a bootstrap approach (1000 iterations), 95% confidence intervals of the mean were estimated. For each subject we computed thresholded onset and peak latencies of the classification time course. For this, we adopted the method from *Marti et al., 2015*. First, the data were low-pass filtered at 10 Hz. Second, to measure the latency of the peak, we identified the first peak within the tROI that was above one or two standard deviations, respectively, of the mean baseline accuracy. If no peak was found, we considered all time points within the tROI that exceeded the 95th percentile of the distribution of the classification performance. Then the median of these points was considered as the peak latency. Finally, from the peak latency, the onset of the peak was defined by going backward and identifying the time point at which the classification performance exceeded a certain threshold percentage of the peak. We performed this analysis with different percentage values (10%, 25%, 30%, 50%) of the difference between the mean decoding accuracy during baseline and the accuracy peak value (results were similar across different threshold values). If the peak latency criteria could not be met for one source, rIFG or pre-SMA, the subject was excluded. Statistical differences between onset latencies of rIFG and pre-SMA at group level were assessed as in the TFR-based latency analysis with permutation statistics (50,000 permutations). To check if the effects we found using broadband data as input for the SVM were specific to beta-band oscillations, we repeated the above described analyses on beta-band filtered data. The source signal was filtered between 12 and 32 Hz using a one-pass Butterworth filter.

## Bayesian multiple regression for SSRT analysis

To analyze how SSRT is influenced by changes in beta-band power or connectivity between rIFG and pre-SAM, we performed a Bayesian multiple regression with per-subject SSRT values as outcome variable and beta power (sSTOP–cAC-GO) in pre-SMA and rIFG in successful and failed trials, as well as the DAI (see below) as regressors. All participants that showed a positive beta power peak during the tROI (as described for the onset definition based on TFRs) in both trial types and both regions were included into the analysis (outliers above three standard deviations were excluded, $n = 43$).

We chose to model the SSRT as normally distributed across subjects, as it is (despite its name) not a directly observable RT but the result of an extraction procedure that involves multiple samples per SSRT value. We $z$-normalized all variables before modeling and used a normal distribution with zero mean and a standard deviation of 0.5 as the prior for each regressor. The likelihood was modeled as a normal distribution centered on the regression model with a standard deviation drawn from a half normal prior with standard deviation of 1. Posterior distributions of the parameters were computed in PyMC3 (*Salvatier et al., 2016*) using the Hamiltonian Monte Carlo No U-Turn sampler (NUTS), with 2000 tuning samples and 20,000 samples for the joint posterior distribution.

We compared three models for their expected posterior log-probability using an LOO CV procedure as suggested in *Gelman et al., 2014*. The three models are described in the 'Results' section.

## Connectivity
### Nonparametric Granger causality

For the computation of cGC, we employed a multivariate nonparametric spectral matrix factorization. We computed the CSD matrix of the source signals on the tROI (100–350 ms) using the fast Fourier transform in combination with multitapers (5 Hz smoothing). Using the nonparametric variant of cGC (*Dhamala et al., 2008*) avoids choosing a multivariate autoregressive model order, which can introduce a bias. Specifically, we used a blockwise approach (*Wang et al., 2007*), considering the first two PCs of each source signal as a block, and we estimated the cGC that a source $X$ exerts over a source $Y$ conditional on the remaining areas (*Bastos et al., 2015*). According to our hypothesis, testing for differences between both trial conditions was done for two links, from rIFG to pre-SMA, and vice versa, conditional on all other five active cortical sources as obtained by the source reconstruction. Even though unobserved sources pose an irresolvable problem (*Bastos and Schoffelen, 2015*), and we cannot totally exclude this scenario, we made use of all the information from the sources observed to distinguish between direct and indirect effects and thus avoiding possible spurious results.

## Statistical testing of cGC

First, we assessed whether the average cGC (in the frequency range of 8–44 Hz) of the source-target pairs (rIFG to pre-SMA and vice versa) was reliably above the bias level, for each condition (sSTOP and cAC-GO) separately. In order to estimate the bias, we randomly permuted the trials 1000 times in each condition to create a surrogate distribution of mean cGC values. We tested if the found cGC value was in the upper 97.5% extreme (equivalent p<0.05 with a Bonferroni correction for two possible source-target pairs) of the surrogates distribution. If the average cGC exceeded the bias level, this source-target link was considered significant. These steps were repeated for each subject separately. Second, for both source-target pairs, cGC values in the sSTOP condition were contrasted with cGC value in the cAC-GO condition at the group level on the subjects that showed at least a significant link in one of the two conditions. The statistical comparison was performed in the range 8–44 Hz using a dependent-samples permutation $t$-metric. A cluster-based correction was used to account for multiple comparisons across frequencies (*Maris and Oostenveld, 2007*). Adjacent frequency samples with uncorrected p-values of 0.05 were considered as clusters. Fifty-thousand permutations were performed and the critical $\alpha$ value was set at 0.025. A Bonferroni correction was applied to account for multiple comparisons across links.

## Directed influence asymmetry index

The DAI was defined by *Bastos et al., 2015* as

$$\mathrm{DAI} = \frac{\mathrm{GC}(A \rightarrow B) - \mathrm{GC}(B \rightarrow A)}{\mathrm{GC}(A \rightarrow B) + \mathrm{GC}(B \rightarrow A)}$$

where in our analysis brain area $A$ was rIFG and brain area $B$ was pre-SMA. The numerator represents the net direction of GC (or cGC, respectively) while the denominator normalizes the net GC by the sum of GC in both directions such that it is possible to compare it across frequencies (*Michalareas et al., 2016*).

## Acknowledgements

This project was funded by German Research Foundation (CRC 1193, project C04, MW and OT). Parts of this research were conducted using the supercomputer Mogon and advisory services offered by Johannes Gutenberg University Mainz (hpc.uni-mainz.de), which is a member of the AHRP (Alliance for High Performance Computing in Rhineland Palatinate, http://www.ahrp.info) and the Gauss Alliance e.V. The authors gratefully acknowledge the computing time granted on the supercomputer Mogon at Johannes Gutenberg University Mainz.

## Additional information

### Funding

| Funder | Grant reference number | Author |
|---|---|---|
| Deutsche Forschungsgemeinschaft | SFB 1193 | Michael Schaum Edoardo Pinzuti Alexandra Sebastian |

The funders had no role in study design, data collection and interpretation, or the decision to submit the work for publication.

### Author contributions

Michael Schaum, Investigation, Formal analysis, Software, Data curation, Resources, Visualization, Writing - original draft, Writing - review and editing; Edoardo Pinzuti, Data curation, Software, Formal analysis, Writing - original draft, Writing - review and editing; Alexandra Sebastian, Conceptualization, Software, Formal analysis, Investigation, Visualization, Writing - original draft, Writing - review and editing; Klaus Lieb, Michael Wibral, Supervision, Funding acquisition, Writing - review and editing; Pascal Fries, Conceptualization, Resources, Supervision, Writing - review and editing; Arian Mobascher, Conceptualization, Writing - review and editing; Patrick Jung, Conceptualization,

Investigation, Writing - review and editing, Supervision; Oliver Tüscher, Conceptualization, Supervision, Writing - original draft, Writing - review and editing, Funding acquisition

### Author ORCIDs
Michael Schaum https://orcid.org/0000-0002-6589-4530
Edoardo Pinzuti https://orcid.org/0000-0002-5113-723X
Alexandra Sebastian http://orcid.org/0000-0002-8381-8312
Pascal Fries https://orcid.org/0000-0002-4270-1468
Patrick Jung https://orcid.org/0000-0003-1242-2844
Michael Wibral https://orcid.org/0000-0001-8010-5862
Oliver Tüscher https://orcid.org/0000-0002-4023-5301

### Ethics

Human subjects: All individual participants included in the study provided written informed consent before participation and consent to publish any research findings based on their provided data in anonymized form. The study was approved by the local ethics committees (Johann Wolfgang Goethe University, Frankfurt, Germany, and Medical Board of Rhineland-Palatinate, Mainz, Germany, IRB Protocol no. 837.128.11), and participants were financially compensated for their time.

### Decision letter and Author response

Decision letter https://doi.org/10.7554/eLife.61679.sa1
Author response https://doi.org/10.7554/eLife.61679.sa2

## Additional files

### Supplementary files
• Transparent reporting form

### Data availability

All data generated or analysed during this study are included in the manuscript and supporting files. All source data files are available on Dryad Digital repository (https://doi.org/10.5061/dryad.x3ffbg7gp). All custom Matlab codes used in these analyses are available at https://github.com/meglab/acSST (copy archived at https://archive.softwareheritage.org/swh:1:rev:ea0bf4acc0f11cdc78ad31b6c1285f1851389312).

The following dataset was generated:

| Author(s) | Year | Dataset title | Dataset URL | Database and Identifier |
| --- | --- | --- | --- | --- |
| Schaum M | 2021 | Right inferior frontal gyrus implements motor inhibitory control via beta-band oscillations in humans | https://doi.org/10.5061/dryad.x3ffbg7gp | Dryad Digital Repository, 10.5061/dryad.x3ffbg7gp |

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
