## [Decision Letter]

**Acceptance summary:**

This manuscript is a valuable addition towards understanding the neural mechanisms of inhibitory control. The combination of MEG with fMRI in a large cohort of subjects using a behavioral paradigm that takes account of attentional confounds, provides a rich dataset in which to determine the temporally primary cortical node in action stopping – the pre-supplementary motor area (pSMA) or right inferior frontal gyrus (rIFG). Here it is shown using a number of techniques, that β activity in the rIFG begins prior and is granger causal to the pSMA, and predicts stopping behavior. As detailed in the discussion, this significantly improves our understanding of this network and provides a platform for consideration of future therapies directed towards rIFG inhibitory β.

**Decision letter after peer review:**

Thank you for submitting your article "Right inferior frontal gyrus implements motor inhibitory control via β-band oscillations in humans" for consideration by *eLife*. Your article has been reviewed by four peer reviewers, including Simon Little as the Reviewing Editor and Reviewer #1, and the evaluation has been overseen by Richard Ivry as the Senior Editor.

The reviewers have discussed the reviews with one another and the Reviewing Editor has drafted this decision to help you prepare a revised submission.

As the editors have judged that your manuscript is of interest, but as described below, see the need for additional analyses and revisions are required before it could be considered for publication. We would like to draw your attention to changes in our revision policy that we have made in response to COVID-19 (https://elifesciences.org/articles/57162). First, because many researchers have temporarily lost access to the labs, we will give authors as much time as they need to submit revised manuscripts. We are also offering, if you choose, to post the manuscript to bioRxiv (if it is not already there) along with this decision letter and a formal designation that the manuscript is "in revision at *eLife*". Please let us know if you would like to pursue this option. (If your work is more suitable for medRxiv, you will need to post the preprint yourself, as the mechanisms for us to do so are still in development.)

This study uses a multimodal neuroimaging approach to investigate the cortical inhibitory control network in a significant number of subjects with a modified version of the stop signal reaction time task. The authors examine the process of stopping a movement with the aim to show that this process is initiated in the right IFG rather than pre-SMA. To this end, they perform MEG source localization analysis of response latencies in the source data and directionality analysis using a variant of Granger causality. The central question that is addressed is whether the IFG is the hub of inhibitory control, or does response inhibition initiate within the SMA. Previous studies have been inconclusive. Some have been limited by poor temporal resolution (fMRI), while recent MEG studies suggest both are active. Another important question and unresolved issue regarding stop inhibition is how to disentangle attentional from inhibitory processes. To address this, this study utilizes attentional capture go trials and compares activity between those trials and stop trials. This allows to partially distinguish between an attentional signal and an inhibitory signal.

The main findings was of a significantly earlier β band response latency in rIFG than pre-SMA and that rIFG β but not pre-SMA β is correlated with stop signal response time. Additional analyses showed a significant correlation for a modified Granger causality from rIFG to pre-SMA but not vice versa and a partially overlapping activation network in the same task using fMRI compared to MEG.

Although there was generally some enthusiasm for the manuscript, writing and the analysis, there was also convergence on a number of concerns across reviewers. Most specifically, it was felt that the main finding, that rIFG precedes pre-SMA needs further evidence to support it. The other findings of rIFG and pre-SMA involvement in stopping support previous findings. The main reviewers concerns are summarized here below and particular attention should be made to addressing these:

The question was repeatedly raised of the use of contrasts in the β power analysis (rather than showing the non-contrasted β amplitudes). Specifically, it is not possible to know if there was a genuine increase in β related to stopping, or simply a slowing of β event related desynchronization (reduction) in the stopping condition compared to the attentional capture condition. This would have important consequences for mechanistic interpretations of the signal and whether it can be truly considered as an initiator of the stopping network, but wasn't clear from the results or figures.

The appearance of low frequency activity in the spectrogram in Figure 3 and the question of whether the IFG – SMA inhibitory effect is also mediated by non – oscillatory evoked activity (which also appears in the low frequency range). This would require a further analysis looking to determine if the effects were truly specific to β or could also be found in non – oscillatory (evoked) activity. Furthermore, multiple reviewers felt that a better mechanistic account was needed of how such short timing effects (e.g. 20 ms difference in the SVM model classification difference) could be driven by β with a cycle length of 50ms and much clearer explanation of why the β frequency ranges were so different across analyses. Notably – the 30-40 Hz signal in the Granger causality analysis does not appear to be the same signal being analyzed in the latency analysis. This either requires further analysis of matching frequencies or significant further explanation and more measured interpretation.

Correspondence between fMRI versus MEG was not felt to be as strong as described in the write up. This raised the question for some reviewers as to whether the regions of interest where truly chosen in a data driven way. A further control analysis of another region such as pre-motor cortex and left IFG was felt to be necessary to ensure that these effects seen are genuinely restricted and specific to the rIFG and pre-SMA in the time period of interest.

Elaboration on the above issues:

1. The author use a comparison of statistics rather than a direct statistical comparison of correlations. This comparison was felt to be invalid by a number of reviewers and a direct statistical comparison was requested (see below for details). Statistics – The argument for the difference between rIFG and pre-SMA (Figure 3D) is based on comparison of statistics rather than a statistic for comparison (see https://www.nature.com/articles/nn.2886). Even ignoring that, the p-value of 0.042 with N=50 does not seem to be very convincing and looking at the figure the two point clouds are not very different. It is unclear how the results will stand up if this essential comparison is done..

2. Statistic and corrections – Correlation analysis: For demonstrating the difference between rIFG and preSMA correlations these must be directly compared to each other rather than showing a significant correlation in one, and the absence of that in the other (please see Nieuwenhuis et al. 2011, Nature Neuroscience). Also, though not relevant when comparing preSMA and rIFG directly, when using multiple correlations these should be corrected for multiple comparisons. As we understand the analyses, four correlation analyses were performed preSMA and IFG for different contrasts (uSTOP and sSTOP). The rIFG correlation seems to only survive when not correcting for multiple comparisons and using a one-tailed p.

3. Contrasts – "he authors almost exclusively report relative changes in MEG activity in one condition (successful stop, sSTOP) compared to another condition (attentional capture GO, cAC-GO). Even though the authors try to take into account that in one condition people pressed a button (cAC-GO) while they do not execute a motor response in the other (sSTOP) by specifying a certain time window, they still cannot control for differences related to the well-known β-decrease during a motor response, which starts several hundred ms before a response. Looking at the median RT, there is presumably strong β-desynchronisation during the time window of interest. So any 'β-increase' in sSTOP vs cAC-GO, which is interpreted as stopping-related, might simply reflect that there is less β decrease in sSTOP compared to cAC-GO, because there is not motor response. In fact, supplementary figure 2 shows that β strongly decreased (when pooled across conditions), and not increased, after the cue / before the response. In order to support the claim that a β increase (and not the lack of a decrease) is related to stopping, the authors would have to show that β in fact increases in sSTOP-GO compared to baseline activity (e.g. before the cue).

4. Contrasts – "Figure 3 and Methods L. 1248 – I was not able to follow the authors' description of the difference between what is shown in 3A and 3B other than that 3A shows t-value and 3B – z-value. When you talk about z-transforming trials, did you z-transform the raw data or the TF power? If it is the former, then the colour in 3B would not correspond to z-value. What makes sense to me is that in both cases you averaged the power across trials for each subject but in 3A looked at differences in power whereas in 3B – at differences in latency but either I am wrong or the description is unclear."

5. fMRI – "Description of Figure 2B in the text implies that there was specific activation of rIFG and preSMA in the fmri study, but the figure shows broad activation including bilateral prefrontal, parietal and occipital areas. Thus, even though the authors state that they did the source reconstruction for analyzing sources based on the current experiment and not based on previous literature picking out preSMA and rIFG for the next analyses seems to be largely related on previous literature and not the current study. For example, how would these analyses look like for left IFG or premotor areas? While I understand the rationale for looking at rIFG more closely based on the current data (due to the early onset of spectral differences), picking out preSMA from the other sources seems to be based on previous literature."

6. fMRI – "In Figure 2 the correspondence between MEG and fMRI results does not seem to be very convincing with the MEG results surprisingly being much clearer. Particularly the absence of activation of bilateral motor strips and pre-SMA in fMRI is puzzling.

a. Would it be possible to quantify the overlap and say how it would compare to other published MEG/fMRI comparisons with the same task?

b. Do you have any explanation for the discrepancies?

c. Also for me it is not obvious that higher β power should necessarily correspond to higher BOLD as at least for movement higher β is associated with movement inhibition. Have you tried looking at the other direction of the fMRI contrast, perhaps the missing sensorimotor cortex is there?

d. Is the IFG latency also significantly different than the latency in the other brain regions in which β band activity is observed?"

7. SVM – "The β latency analysis is supported by the SVM analysis which uses a wide-band frequency input. To strengthen the case that the β band is mechanistically primary here, it would be useful to repeat the SVM analysis using a β filtered input to the SVM. This would confirm that the effects seen in the SVM are driven by the β band.

8. SVM – "Just as a point of clarification, it appears that the onset analysis of the classification reveals a much later time of onset. It would be helpful to understand why this classification time course appears slower than the time course of β power, and whether the difference in classification between IFG and SMA should be interpreted in the same way as the difference in power latency."

9. SVM – The authors report that the SVM was point-by-point. One reviewer commented "I don't understand what that means because the whole idea of SVM is that it's multivariate so could possibly be sensitive to things like β. Point-by-point would mean that somehow the raw signal amplitude at each point is different between trial types so unless there is an evoked response there, I don't see how this would be the case. If it's really point-by-point the latency difference with β power would make sense because the time resolution of TF analysis is limited, in this case to 150ms windows around 20Hz."

10. Frequency of interest – "There is an early 'relative' β-increase in rIFG, which already starts ~100ms after the cue. Its spectral components seem quite different from the later occurring β change and it includes a low-frequency component. Did the authors assess whether this could be an evoked response with different frequency components in the spectral domain? The cluster's duration seems <100 ms which would correspond to less than 2 cycles of a 20Hz oscillation. Also, would this cluster be significant when not 'spilling over' into the later β-cluster and the low-frequency cluster?"

11. Frequency of interest – "Recent evidence, already cited in this manuscript (Chen et al. 2020) suggests that an earlier evoked response from IFG might precede β changes during reactive stopping. The Figure 3A low frequency activity also supports this. Given the SVM changes on broadband data, the findings in Figure 3A and the previous literature, it would be informative if the authors could analyze if evoked (non – oscillatory +- very low frequency δ which is likely equivalent) activity is also earlier (and potentially) causal in IFG  pre-SMA."

12. Frequency of interest – "Much of the discussion and the initial results regarding spectral power are focused on the broad β band from 12-30 Hz. Yet the analysis of Granger causality and DAI are focused on a slightly higher frequency range, 30-40 Hz, which is not usually considered as β when discussing the stopping nature of IFG (in fact, some would consider this in the range of low γ). How should one reconcile these different frequencies?

a. Can the authors re-analyze in matched frequency bands and discuss the non – concordance in frequency given.

b. Moreover, the relation between DAI and SSRT is specific to 34 Hz. I understand that this has been corrected for multiple comparisons, but does this correction just account for the six discrete frequencies between 30-40 Hz, or does this account for the entire β band (or even all frequencies for that matter)? "

13. Frequency of interest – "This relates to interpretation – could the authors discuss and reference how an oscillation with a cycle length ~ 50ms could causally impact one region in the very short time frames demonstrated in this study (e.g SVM difference is 20ms in Figure 3C, ie less than half a cycle)."

14. Granger causality – "Figure 4 – why does Granger causality go down with frequency? I would expect it to be flat except where there is a significant relation between signals. Could it be some kind of power confound or normalisation issue?"

15. DAI – "L. 475 – Was the correlation of DAI with SSRT independent of the IFG power correlation i.e. was the DAI regressor significant in the presence of rIFG power? If not, perhaps the DAI result could be explained by varying rIFG SNR that is known to confound directional measures."

16. Clusters – "L. 1152 – the description of the cluster test seems to be inaccurate. If you took cluster sizes below 0.5th percentile of the distribution that would mean that you took very small clusters as well as very large. What usually happens is that people take large clusters for both positive and negative direction and apply Bonferroni correction for the two directions. Also why was a conservative threshold used p<0.01? Would there be additional clusters for p<0.05?"

17. Code – "Although the authors included links to the data and the custom code in the cover letter, they do not appear in the paper itself. It would be helpful to include a supplementary table pointing to the toolboxes and custom functions used for each analysis. I was wondering particularly about cGC as no reference for the implementation was given and the conditional part sounds like more than is implemented in Fieldtrip."

[Editors' note: further revisions were suggested prior to acceptance, as described below.]

Thank you for resubmitting your work entitled "Right inferior frontal gyrus implements motor inhibitory control via β-band oscillations in humans" for further consideration by *eLife*. Your revised article has been evaluated by Richard Ivry (Senior Editor) and a Reviewing Editor.

We appreciate your resubmission along with your thorough responses to the suggestions from our reviewers. Many of the issues have been addressed eg. comparison of statistics with your new modelling approach, clarification around ROIs and fMRI with the new analysis of the contract between the two neuroimaging approaches, SVM methodological expansion, frequencies of interest, including evoked responses and others. Overall, this is now more compelling and a stronger paper.

A remaining issue relates to the new plots in Figure 3 and also the interpretation of the β signal. The new plots showing spectral activity per condition are welcome and confirm that there is indeed a reduced β desynchronization in the Stop condition. However, with the spectral (color) plots it is still difficult to accurately determine the time course of the β activity, which here is particularly important given the mechanistic interpretations of β and was a concern of reviewers.

It would be advisable to include one further plot of a time series of the β filtered power in each condition separately as a separate time series. To put it another way – to show a line plot of the time resolved β power (rather than a spectrogram) with error bars. This would give a clearer view of whether the β plateaus, slows its desynchronization, slightly increases or whether the variance increases, all of which could be driving change in significance and are difficult to determine from the spectrogram. This is important for the mechanistic interpretation and particularly for the "Inhibitory role of β-band oscillations" Discussion section.

While there is agreement that the study "suggests an active change of processing in rIFG that is related to stopping, while processing proceeds 'as usual' in cAC-GO trials", the role of β here and whether this it has a primary stopping function or these changes are secondary to a different mechanism for stopping in the IFG is still undetermined here. The current Figure 3 and your aligning with the status quo theory suggests that β changes here are more secondary and related to a withdrawal of an ongoing action initiation process rather than new active stopping process. It should be discussed further and potentially highlighted that a reduction of an ongoing motor initiation process (e.g. reducing β ERD) is different from a specific, distinct and active "stopping" process.

Your current discussion was at times still felt to be implying that β was mechanistically a stopping signal (which would have been supported by an early increase in β with stopping) rather the possibility that β desync reflects motor initiation which was arrested (as implied by a slowing or plateauing or the ERD). An analogy might be the different between reducing pressure on an accelerator pedal, versus actively pressing a brake. In general this should be addressed throughout and the following are particular example areas of text which could be clarified to take account of this β discussion above:

"We suggest that these differences then result in the observed net β-band power increase before SSRT". – This implies β increases, whereas the data thus far presented doesn't support this.

"Inhibitory role of β oscillations" – This isn't definitively shown here.

"Our results strongly support the interpretation that the rIFG acts as the initiator of response inhibition and that β-band oscillations play an important role for its neural implementation." – This implies β is mechanistic for response inhibition – but again this isn't shown here definitively, and β changes could secondarily result from a pausing of action initiation.

Finally – whilst Figure 3 is a welcome addition – the curved greyed out area suggests a strong windowing effect from Time Frequency (TF) decomposition on short time window / epochs causing distortion at the edges which are worse for low frequencies (which have been masked out). This is unconventional to analyse and present like this, with normal practice being to perform the TF decomposition on the continuous data and to epoch power estimates afterwards, or to perform TF decomposition on a much wider epoch of data per trial and then re-epoch / chop down to the (same) time period of interest (e.g. -0.2 to 0.5). It is advised to repeat this analysis with better epoching – currently the windowing is artificially constraining the time and frequencies analysed.

---

## [Author Response]

[…]The appearance of low frequency activity in the spectrogram in Figure 3 and the question of whether the IFG – SMA inhibitory effect is also mediated by non – oscillatory evoked activity (which also appears in the low frequency range). This would require a further analysis looking to determine if the effects were truly specific to β or could also be found in non – oscillatory (evoked) activity. Furthermore, multiple reviewers felt that a better mechanistic account was needed of how such short timing effects (e.g. 20ms difference in the SVM model classification difference) could be driven by β with a cycle length of 50ms and much clearer explanation of why the β frequency ranges were so different across analyses. Notably – the 30-40 Hz signal in the Granger causality analysis does not appear to be the same signal being analyzed in the latency analysis. This either requires further analysis of matching frequencies or significant further explanation and more measured interpretation.Correspondence between fMRI versus MEG was not felt to be as strong as described in the write up. This raised the question for some reviewers as to whether the regions of interest where truly chosen in a data driven way. A further control analysis of another region such as pre-motor cortex and left IFG was felt to be necessary to ensure that these effects seen are genuinely restricted and specific to the rIFG and pre-SMA in the time period of interest.Elaboration on the above issues:1. The author use a comparison of statistics rather than a direct statistical comparison of correlations. This comparison was felt to be invalid by a number of reviewers and a direct statistical comparison was requested (see below for details). Statistics – The argument for the difference between rIFG and pre-SMA (Figure 3D) is based on comparison of statistics rather than a statistic for comparison (see https://www.nature.com/articles/nn.2886). Even ignoring that, the p-value of 0.042 with N=50 does not seem to be very convincing and looking at the figure the two point clouds are not very different. It is unclear how the results will stand up if this essential comparison is done..

We agree with the above arguments. To understand the effects of changes in β power on the stop signal reaction time (SSRT) in more detail we performed a Bayesian multiple regression analysis with per subject SSRT-values as outcome variable and β power in presupplementary motor area (pre-SMA) and right inferior frontal gyrus (rIFG) in successful and failed trials, as well as the directionality asymmetry index (DAI) as regressors (please refer to question 15, which was related to correlation with SSRT, too).

We chose to model the SSRT as normally distributed across subjects, as it is (despite its name) not a directly observable reaction time but the result of an extraction procedure that involves multiple samples per SSRT-value. We *z*-normalized all variables before modelling and used a normal distribution with zero mean and a standard deviation of 0.5 as a the prior for each regressor. The likelihood was modeled as a normal distribution centered on the regression model with a standard deviation drawn from a half normal prior with standard deviation of 1. Posterior distributions of the parameters were computed in PyMC3 (Salvatier et al., 2016) using the Hamiltonian Monte Carlo No U-Turn sampler (NUTS), with 2000 tuning samples and 20.000 samples for the joint posterior distribution.

We compared three models for their expected posterior log-probability using a leave-one-out cross-validation (LOO-CV) procedure as suggested in Gelman et al. (2014). Model 1 used all regressor variables and their first order interaction terms, model 2 used all regressor variables, but no interaction terms, and model 3 additionally averaged together the β power in pre-SMA for successful and failed trials as the values of these regressors were highly correlated (*r* > 0.6). Thus, model complexity was reduced successively from model 1 to model 3. Model 3 achieved the best LOO-CV score (expected log posterior predictive density elpdloo = –64.65 ± 5.32, higher is better, maximum is at zero), but model 2 was within one standard error of model 3 elpdloo = –65.24 ± 5.10. Model 1 had a considerably worse LOO-CV score (elpdloo = –71.67 _ 4.53).

In the winning model 3, there was as 91.0 % marginal posterior probability for a negative regression coefficient of the β power in rIFG for successful trials, and a 87 % marginal posterior probability for a negative regression coefficient for the DAI, but only a 64 % marginal posterior probability for a negative regression coefficient for the average β power in pre-SMA, and a 40 % marginal posterior probability for a negative regression coefficient for the β power in IFG in failed trials.

For model 2, with individual regressors for β power in the pre-SMA, results were highly similar, with a 89 % marginal posterior probability for a negative regression coefficient for β power in rIFG in successful trials, and a 88 % marginal posterior probability for a negative regression coefficient for the DAI, with all other marginal distributions of regression coefficients being more symmetric around 0.

Given these results, we consider it most plausible that β power in rIFG in successful trials and the DAI predict SSRT outcomes, with all other regressors having no or a much smaller influence. We revised the respective paragraph (results: l. 362 ff, methods: l. 1519 ff) accordingly.

This new analysis supports the claims made previously on the role of β-band power in rIFG for stopping, and replaces the initial analysis which was indeed insufficient.

2. Statistic and corrections – Correlation analysis: For demonstrating the difference between rIFG and preSMA correlations these must be directly compared to each other rather than showing a significant correlation in one, and the absence of that in the other (please see Nieuwenhuis et al. 2011, Nature Neuroscience). Also, though not relevant when comparing preSMA and rIFG directly, when using multiple correlations these should be corrected for multiple comparisons. As we understand the analyses, four correlation analyses were performed preSMA and IFG for different contrasts (uSTOP and sSTOP). The rIFG correlation seems to only survive when not correcting for multiple comparisons and using a one-tailed p.

We agree with this point. Therefore, we decided to use a full Bayesian analysis as described above to address this issue.

3. Contrasts – "he authors almost exclusively report relative changes in MEG activity in one condition (successful stop, sSTOP) compared to another condition (attentional capture GO, cAC-GO). Even though the authors try to take into account that in one condition people pressed a button (cAC-GO) while they do not execute a motor response in the other (sSTOP) by specifying a certain time window, they still cannot control for differences related to the well-known β-decrease during a motor response, which starts several hundred ms before a response. Looking at the median RT, there is presumably strong β-desynchronisation during the time window of interest. So any 'β-increase' in sSTOP vs cAC-GO, which is interpreted as stopping-related, might simply reflect that there is less β decrease in sSTOP compared to cAC-GO, because there is not motor response. In fact, supplementary figure 2 shows that β strongly decreased (when pooled across conditions), and not increased, after the cue / before the response. In order to support the claim that a β increase (and not the lack of a decrease) is related to stopping, the authors would have to show that β in fact increases in sSTOP-GO compared to baseline activity (e.g. before the cue).

We agree with your observation that the reported increases in β power are relative and not absolute increases. We have now added the contrasts (“main effects”) for task vs. baseline to Figure 3 in the revised manuscript (TFRs for both conditions, sSTOP and cAC-GO). Nonetheless, those contrasts support our interpretation of a specific, though relative, increase in β power for the stop condition, when compared side by side with the cAC-GO condition. As you pointed out, β-band desynchronization (BBD) is related to motor-response preparation and BBD already sets in early after the initial GO signal. And indeed, until approximately 100 ms after the STOP/AC-GO signal occurs, we observe almost the same strength of BBD for both conditions (see Figure 3, panels A, B in the revised manuscript). This is confirmed by the contrast between both conditions showing that BBD almost cancels out in this time window (see Figure 3, panel C in the revised manuscript). While in cAC-GO trials, the BBD continues until and beyond the median RT (first dashed line in Figure 3), BBD is aborted in sSTOP trials already approximately 100–150 ms before the median SSRT (first solid line in Figure 3). Hence, from the perspective of an already initiated BBD, what we found is not just a lack of BBD after the stop signal compared to cAC-GO trials, but an early *termination* of BBD in sSTOP trials within the time window of interest. This means that our results do not just originate from the contrast of sSTOP with cAC-GO trials, but the BBD termination sets in already before the SSRT in the sSTOP condition, while in the cAC-GO trials no such termination was observed. This finding suggests an active change of processing in rIFG that is related to stopping, while processing proceeds ’as usual’ in cAC-GO trials, and the motor response will be executed. Yet, we didn’t find an absolute β-band increase *before* SSRT in sSTOP trials compared to baseline, which is in line with a recent ECoG study based on 21 patients (Chen et al. (2020), see Figure 4B, Figure S6). Thus, taken together for the results that relied on the contrast between the conditions sSTOP and cAC-GO, as well as their single contrast components (task-versus-baseline TFRs), we found clear evidence that the relative change of β-band oscillations specifically within the sSTOP condition plays a crucial role in response inhibition. We revised the respective paragraph in the discussion (l. 635 ff) accordingly.

4. Contrasts – "Figure 3 and Methods L. 1248 – I was not able to follow the authors' description of the difference between what is shown in 3A and 3B other than that 3A shows t-value and 3B – z-value. When you talk about z-transforming trials, did you z-transform the raw data or the TF power? If it is the former, then the colour in 3B would not correspond to z-value. What makes sense to me is that in both cases you averaged the power across trials for each subject but in 3A looked at differences in power whereas in 3B – at differences in latency but either I am wrong or the description is unclear."

We apologize for the confusion that Figure 3A/B created. Both subfigures rely on the same data. While 3A shows the result of a statistical test that asks for differences between both conditions (non-significant areas outside the clusters were displayed transparent), 3B shows the result of simply subtracting power of cAC-GO from power of sSTOP trials, i. e. it is closer to the raw data.

The data were processed as follows:

1. Trial-wise time-frequency analysis for each subject and condition (task and baseline segments for sSTOP and cAC-GO conditions).

2. *z*-transformation of these single trials (in the frequency domain, not the raw data), by trial-wise baseline correction (standard deviation of the baseline of both conditions pooled was used).

3. Averaging all *z*-transformed trials per subject. For the statistics (Figure 3A), the average over trials of each subject and condition was used as input for the cluster-based permutation test (sSTOP vs. cAC). For the simple difference shown in Figure 3B (sSTOP – cAC), we grand-averaged both conditions across subjects.

So why did we show both representations? The appearance of the early significant cluster in rIFG (Figure 3A) simply states that there indeed exists an early difference between both conditions. However, cluster-based permutation tests do not allow a precise statement about the onset of a certain effect (Sassenhagen and Draschkow, 2019), as cluster border locations come without statistical guarantees of any kind (just cluster existence is statistically guaranteed). Since we aimed to test our hypothesis for differences between the onset timing in rIFG and pre-SMA, we therefore used averaged, *z*-transformed TFR data obtained by step 3 as described above. With Figure 3B we aimed to highlight these different approaches but given the confusion we removed it and added the single condition TFRs (task vs. baseline) instead (see point 1). Additionally, we improved the corresponding section in the methods section (l. 1418 ff). Thus, we hope that the comprehensibility of the manuscript was improved.

5. fMRI – "Description of Figure 2B in the text implies that there was specific activation of rIFG and preSMA in the fmri study, but the figure shows broad activation including bilateral prefrontal, parietal and occipital areas. Thus, even though the authors state that they did the source reconstruction for analyzing sources based on the current experiment and not based on previous literature picking out preSMA and rIFG for the next analyses seems to be largely related on previous literature and not the current study. For example, how would these analyses look like for left IFG or premotor areas? While I understand the rationale for looking at rIFG more closely based on the current data (due to the early onset of spectral differences), picking out preSMA from the other sources seems to be based on previous literature."

It was not our intention to give the impression that only rIFG and pre-SMA activity were found in the fMRI contrast. We revised the manuscript to avoid this impression (l. 239 ff). In the original manuscript we had reduced the comparison to the specific regions we were interested in, based on our hypotheses. We also would like to clarify that the overall approach of this study was highly hypothesis driven with the purpose to investigate timing and role of rIFG and pre-SMA. This is because, both, pre-SMA and rIFG, represent the key areas involved in response inhibition as shown by many previous studies. In other words, the question “Who is first, pre-SMA or rIFG?” has crystallized over many previous studies, but had not been resolved before, hence our strong focus on this hypothesis/comparison.

The term “data-driven” referred to the identification of *coordinates* of sources at which oscillatory activity can be actually observed in the acquired MEG data. This is in contrast to a literature-based selection of coordinates. The word “coordinates” was inserted in the misleading sentence (l 191 ff.) to emphasize this approach. Many former MEG studies a priori defined anatomical ROIs and their coordinates (or masks) instead and extracted data using virtual channels without testing or at least without reporting whether these locations actually revealed significant activity during response inhibition given their paradigm and measurement data or not. This seed-based approach, however, is explicitly discouraged in the current best practice guidelines for MEG analyses (Gross et al., 2013). The recommended approach of contrasting appropriate conditions that we used enabled us not only to identify coordinates of peak activity for rIFG and pre-SMA, i. e. the location with the strongest effect across subjects, but also excluded that the effects we analyzed were due to leakage from other sources. Indeed, other sources were active at the same time window, too, and we decided to present them in the figure as well to provide a better view of the overall activity. Note that rIFG and pre-SMA were also revealed by the contrast using a smaller (cluster) _-value of 0.01, while other sources would not have survived this threshold (like l-IFG, lAIns, Fornix). Still, the overall approach of this study was highly-hypothesis driven and aimed at revealing timing and roles specifically of rIFG and pre-SMA during response inhibition. Both key areas could be localized quite accurately using fMRI and MEG (based on the same task), and within close distance to each other – as revealed by the comparison between both methods.

6. fMRI – "In Figure 2 the correspondence between MEG and fMRI results does not seem to be very convincing with the MEG results surprisingly being much clearer. Particularly the absence of activation of bilateral motor strips and pre-SMA in fMRI is puzzling.a. Would it be possible to quantify the overlap and say how it would compare to other published MEG/fMRI comparisons with the same task?

In order to compare the source activation between both methods, MEG and fMRI, we created another plot that shows the overlap (see Figure 2C in the revised manuscript). The overlap was defined by the product of absolute, normalized *t*-values for each voxel *i*, |*ti*,MEG| _ |*ti*,fMRI| (MEG data were interpolated to match the fMRI resolution). Normalization was performed for each method separately by division with the maximum *t*-value obtained by the respective method, resulting in normalized *t*-values between 0 and 1.

Comparing the sources revealed by the MEG and fMRI contrasts showed an overall good correspondence. Given the 1 cm grid used for the MEG source reconstruction, the distance between peaks of both methods were rather close for rIFG (7 mm), pre-SMA (15 mm), lAIns (11 mm), and lMFG (15 mm). The peaks of premotor areas (rPMC, lPMC) as well as l-IFG were less close to the fMRI results (20–24 mm difference between methods). However, particularly the sensorimotor areas revealed a rather wide-spread distribution of source activation and referring just to the MEG peak voxel may be over-conservative. Yet for the key areas that were most important for this study, rIFG and pre-SMA, we found a very good correspondence and thus could validate that the MEG source reconstruction was quite accurate for these areas. We revised the respective paragraph (l. 248 ff) accordingly.

So far, to our best knowledge, there are no other combined MEG/fMRI studies yet that used a stop signal task. However, if we receive a hint with a reference, we will be happy to include it in the manuscript and also discuss the comparison.

b. Do you have any explanation for the discrepancies?

We agree that the absence of bilateral motor strips in fMRI was confusing. Since we were focused on the comparison of rIFG and pre-SMA for successful stopping (i.e., sSTOP > cAC-GO) between both methods, we didn’t show the fMRI contrast cAC-GO > sSTOP in the original manuscript (also see next point, c.). Now we included this contrast in Figure 2B in the revised manuscript and it can be seen that sensorimotor cortex shows activation, too. However, we disagree that pre-SMA activation was absent in the fMRI analysis. In the Results section, we reported the exact coordinates of pre-SMA activation revealed by the contrast (l. 211f., original manuscript). Yet, due to the surface projection, it was not fully visible. Thus, discrepancies should be eliminated now (see also discussion in a./c.).

c. Also for me it is not obvious that higher β power should necessarily correspond to higher BOLD as at least for movement higher β is associated with movement inhibition. Have you tried looking at the other direction of the fMRI contrast, perhaps the missing sensorimotor cortex is there?

Indeed, when looking at the fMRI contrast cAC-GO > sSTOP, the sensorimotor cortex is active then as expected, also see b. However, please note that the sign is different compared to the MEG β-band activity. Since the BOLD response cannot be limited to the same tROI as in the MEG, the motor response (button presses) in cAC-GO condition might dominate the contrast in the fMRI activity map.

d. Is the IFG latency also significantly different than the latency in the other brain regions in which β band activity is observed?"

Our original research question was to find out which of the two key areas, rIFG and pre-SMA, comes first during response inhibition. The limitation to analyze only these two brain regions was highly hypothesis driven (see also point 5.). For the sake of completeness, we presented all sources obtained by contrasting sSTOP and cAC-GO trials during the tROI. Indeed, rIFG and pre-SMA were active as hypothesized and allowed us to obtain the most accurate (data-driven) coordinates for our analysis. The analysis of all other sources would be just exploratory, and the same applies to the connectivity analysis. Still, it is interesting to see that even at this level, rIFG seems to be the dominant sender within the motor/inhibitory network (see Author response image 1). However, these results were not significant when correcting for multiple comparisons (42 links in total).

The mean onset latency of rIFG was indeed the earliest compared to all other sources. For a statistical comparison of the latencies of all brain regions in which β band activity was observed, we additionally computed a repeated measures ANOVA. Since onset latency was defined (roughly) by a relative positive peak, subjects that did not show such a peak at least in one of the seven sources were excluded (to be consistent with the latency-statistics reported in the original manuscript). Thus, the number of subjects still included would decrease to *n* = 40 and only the latency difference between rIFG and lAI was found to be significant by post hoc tests. However, we would like to point out that interindividual differences in the neural response of stopping in the rIFG and in the β band may vary a lot (see, e. g. in Wessel et al. (2013), Figure 2, Swann et al. (2009), Figure 5). This said, statistical power for the latency analysis of all sources might be too low. Yet, our study was strongly focused on revealing the timing specifically of rIFG and pre-SMA during response inhibition, and we clearly found a temporal precedence of rIFG over pre-SMA activation.

7. SVM – "The β latency analysis is supported by the SVM analysis which uses a wide-band frequency input. To strengthen the case that the β band is mechanistically primary here, it would be useful to repeat the SVM analysis using a β filtered input to the SVM. This would confirm that the effects seen in the SVM are driven by the β band.

We would like to thank the reviewer for suggesting this additional analysis. In the original analysis, we used broadband filtered data for the SVM analysis. Yet, the result of temporal precedence of rIFG over pre-SMA as found for the TFR analysis was confirmed. To check if the effect revealed by the SVM broadband analysis was β-band driven, we additionally performed the same SVM analysis on β-band filtered data (still in the time-domain), but with time-embedded vectors as input as suggested in point 9. Then the above-chance level classification was significant in rIFG at 132 ms (*p* < 0.0001), and in pre-SMA at 142 ms (*p* < 0.0001), tested between 100–350 ms. We added these results as additional plot in Figure 4 in the revised manuscript (panel B) and updated the respective paragraph (l. 335 ff) accordingly.

These findings are close to the one of the broadband analysis (rIFG at 129 ms, pre-SMA at 163 ms) and support the timing of the TFR analysis. However, we observed that the accuracy of the β-band filtered analysis decreased compared to the broadband analysis. We guess that the reduction of accuracy might be due to the missing contribution of evoked responses (as indicated by a reviewer in point 9). This reduction was in particular found after the median SSRT. Since we found significant differences between conditions in the evoked response also after SSRT (Author response image 4 , we guess that indeed evoked responses are crucial for the high classification accuracy after the SSRT, that decreased in the β-filtered compared to the broadband SVM analysis. Yet, before SSRT, we found an early onset of significant above-chance classification when using β-band filtered data, indicating that β-band activity may indeed play an important role for the initiation of response inhibition in the rIFG. However, β-band filtering reduced the power of the signal and individual accuracy curves became quite noisy. Thus, individual onset definition would result in too many exclusions and we were not able to perform the same individual onset statistics as for the broadband data. Hence we could not provide strong support for the temporal precedence of rIFG over pre-SMA when using β-band filtered data. Nonetheless, since the additional SVM analysis on β-band filtered data still revealed an early above-chance classification onset in the rIFG, we conclude that β-band oscillations are carrying the crucial information to reliably classify sSTOP trials and hence, play a leading role in initiating response inhibition in rIFG.

**Author response image 1. sa2fig1:** Exploratory connectivity analysis during the tROI. Spectral-resolved conditional Granger causality (cGC) between all source target pairs (links) was analyzed. Blue curve: cGC for sSTOP trials, green curve: cGC for cAC-GO trials, bounded lines: standard error of the mean. There were no significant differences between both conditions in the frequency range of 8-44Hz when correcting for all 42 links. However, significant differences were found if correction for multiple comparisons was reduced to six links (highlighted in yellow). Uncorrected significance is highlighted in grey.

8. SVM – "Just as a point of clarification, it appears that the onset analysis of the classification reveals a much later time of onset. It would be helpful to understand why this classification time course appears slower than the time course of β power, and whether the difference in classification between IFG and SMA should be interpreted in the same way as the difference in power latency."

The discrepancy can be explained by the differences between the two methods employed. First, the onset values revealed by the latency analysis using TFR as well as the SVM approach depend on the thresholds applied with respect to the peak latency, i. e., higher thresholds would result in later onset values, and the peak characteristics might differ between both analyses. Second, in contrast to the TFR onset analysis, the SVM accuracy curves were low-pass filtered to reduce noise before determining the onset. We therefore believe that the later time onset is due to this filtering step in the analysis pipeline. To check this, we performed an additional analyses using a jacknife approach (Miller et al., 2009) that did not rely on low-pass filtered accuracy curves. In consistency with the SVM analysis of the revised manuscript, we used time-embedded data as SVM input (see questions 7 and 9). This analysis confirmed a mean IFG onset of 130 ms (with low-pass filtering: 178 ms) and a mean pre-SMA onset of 178 ms (with low-pass filtering: 200 ms), *p* = 0.003. Thus, we could confirm that the slower latency values of the individual onset SVM statistics compared to the TFR onset analysis might be due the low-pass filtering for onset definition. More importantly, at least for the rIFG, the onset value found using unfiltered data in combination with the Jacknife approach matched quite well with the onsets found by the TFR analysis, revealing a substantial agreement between both methods (difference was only 7 ms). However, we are aware that the particular method employed might result in slight shifts of the onset values. Yet, our main finding was that the SVM analysis revealed additional evidence for the temporal precedence of rIFG over pre-SMA during response inhibition, and this was found independently of the method used.

9. SVM – The authors report that the SVM was point-by-point. One reviewer commented "I don't understand what that means because the whole idea of SVM is that it's multivariate so could possibly be sensitive to things like β. Point-by-point would mean that somehow the raw signal amplitude at each point is different between trial types so unless there is an evoked response there, I don't see how this would be the case. If it's really point-by-point the latency difference with β power would make sense because the time resolution of TF analysis is limited, in this case to 150ms windows around 20Hz."

In the original analysis we indeed employed a univariate analysis and the classification was done point-by-point, resulting in a linear discriminant analysis in the end. Hence, it’s likely that an evoked response was involved in the effects we found. To be sensitive in particular to β-band oscillations, we performed an additional analysis on β-band filtered data using time-embedding, as suggested in point 7. Using time-embedded vectors as input in general increased the classification accuracy. Hence, we updated the manuscript with results revealed by this multivariate approach (Figure 4A in the revised manuscript). However, even when using a time-embedded multivariate approach, the β-band filtered analysis resulted in lower classification accuracies compared to the broadband analysis (see point 7), which is obviously more heavily influenced by evoked responses. This could be interpreted in a way that successful stopping is also supported by evoked responses. Nevertheless, the early above-chance classification in rIFG was found in the β-band filtered analysis as well. Thus, evoked responses do not seem to be required for the classification of successful stop trials, and most importantly both SVM analyses (using broadband data as well as β-band filtered) confirmed our onset analysis based on TFRs.

10. Frequency of interest – "There is an early 'relative' β-increase in rIFG, which already starts ~100ms after the cue. Its spectral components seem quite different from the later occurring β change and it includes a low-frequency component. Did the authors assess whether this could be an evoked response with different frequency components in the spectral domain? The cluster's duration seems <100 ms which would correspond to less than 2 cycles of a 20Hz oscillation. Also, would this cluster be significant when not 'spilling over' into the later β-cluster and the low-frequency cluster?"

We agree with the reviewer that both questions should be addressed. First, to ensure that the early relative β-increase in rIFG is not just a spectral component of an evoked response, we computed the ERF of the virtual channel and then created a TFR out of the averaged data. Thus, we could identify the spectral components of the ERF (Author response image 2, B). However, when contrasting both conditions as done with the original TFR analysis, we could not find a significant difference in the rIFG (Author response image 2). Hence, we could not find evidence that the early β-band cluster at the rIFG revealed by the time-frequency contrast could be related to evoked-like activity.

Second, we tried to exclude that the early cluster was just a spilling over from other clusters like the low-frequency cluster that was found within the same time range and the later β-rebound cluster (Author response image 3). Note that in contrast to the original manuscript, this new figure shows the TFR statistics performed between 2–44 Hz (as suggested in point 11). In this reanalysis the low-frequency cluster was not significant anymore for the contrast of interest. However, a larger cluster like the later β-rebound would survive permutation-based statistics more easily than a smaller, separate cluster, potentially masking other effects. Hence, we additionally tested in a smaller time-frequency window around the early cluster (100–200 ms, 12–35 Hz) and found that the cluster was still there (Author response image 3). Thus, it seems to be related to a distinct neural process. We interpret the different frequency composition of the early and later relative β-increase in a way that the later β-band increase (after median SSRT) likely reflects the well-known β-rebound effect, while the earlier increase might be specific to the initiation of response inhibition.

In conclusion the additionally performed analyses support our interpretation that early relative β-increase in rIFG represents a distinct oscillatory neural correlate that is related to the initiation of response inhibition and is not related to spectral components of an evoked response.

**Author response image 2. sa2fig2:** Time-frequency representations of evoked responses in rIFG. First and second solid line indicate SSRT and SSRT_max_, first and second dotted line indicate 10% and 50%=percentiles of RT_AC-GO_ for selected AC-GO trials with RT_AC-GO_ > SSRT (RT_AC-GO_ is the duration between AC-GO signal and button press). Time axis locked to STOP and AC-GO signal (Os). (**A**) To obtain evoked responses, successful stop (sSTOP) trials were averaged (for baseline correction, the mean of a 500ms long baseline segment that ended 100ms before the Go signal was used, no low-pass filter applied) and then transformed to TRFs. (**B**) Same as in (A), but for correct attentional capture go trials (cAC-GO). (**C**) Contrast of sSTOP versus cAC-GO trials cluster-based permutation test, tow-tailed, α_cluster_ = 0.05, n=59, no significant clusters found

11. Frequency of interest – "Recent evidence, already cited in this manuscript (Chen et al. 2020) suggests that an earlier evoked response from IFG might precede β changes during reactive stopping. The Figure 3A low frequency activity also supports this. Given the SVM changes on broadband data, the findings in Figure 3A and the previous literature, it would be informative if the authors could analyze if evoked (non – oscillatory +- very low frequency δ which is likely equivalent) activity is also earlier (and potentially) causal in IFG  pre-SMA."

Although Chen et al. (2020) found an early evoked potential in rIFG, it remains unclear whether this EP was specific to stopping or not and whether it set in before the early β changes we found. There are three issues to consider in this context. First, since the results of that study rely on a simple SST, not a selective SST, it was not possible to compare them with an approriate control condition. Second, Chen et al. (2020) showed that IFG–STN *interaction* is mediated primarily by a fast EP rather than by oscillatory synchronization. Yet, the onset of the EP was compared to a β increase that was observed *after* SSRT. As mentioned in our discussion, the β increase after SSRT may reflect a rebound of activity related to action termination, not the actual initiation of response inhibition. From this perspective it seems quite likely that the EP set in earlier than the increase of the late β power due the rebound. Thus it remains an open question whether their EP effect precedes our early β-band effect. Third, Chen et al. (2020) showed that cross-correlation between the EPs from rIFG and STN correlated with SSRT, revealing an important function of the hyper-direct pathway for stopping, but it was not the rIFG EP alone that could be related to stopping. Hence, although the cited study revealed several remarkable results, it should be considered with care when making conclusion about the relationship between evoked and oscillatory responses during the initiation of response inhibition. However, we agree with the reviewer that evoked responses and very low frequency δ oscillations should be analyzed in addition to the oscillatory results we presented.

First, to analyze evoked responses, we generated event-related fields (ERFs) from virtual channel time course data. Since these data originate from a principal component analysis (PCA), signs of the ERFs may be flipped randomly across subjects. To correct for this, we applied the method described by Vidaurre et al. (2018). In this analysis we did not find any differences between conditions before the SSRT at rIFG and pre-SMA (Author response image 4). Second, to analyze very low frequency oscillations, we created TFRs starting at 2 Hz and using a 500 ms time window. We couldn’t find any significant differences between conditions for low frequency activity either (Author response image 4). Also note that for point 10., we checked whether effects in the low-frequency components of the TFRs could originate from effects in evoked responses, which could not be confirmed. Hence, when using the contrast with the cAC-GO control condition, neither effects in evoked responses nor in very low frequency oscillations were setting in already before the early relative β-band increase were found in rIFG. Yet, since absence of evidence is no evidence for absence, we cannot exclude that evoked responses contribute to the initiation of response inhibition. However, in our high-powered study we could not find any evidence that evoked responses or very low frequency oscillations play a crucial role for the initiation of response inhibition.

**Author response image 3. sa2fig3:** Time-frequency statistics focused on early β-band cluster in rIFG. Contrast of *z*-transformed sSTOP versus cAC-GO trials (cluster-based permutation test, two-tailed, α_cluster_ = 0.05, n=59, significant cluster is outlined). (**A**) Tested between -200-500ms and 2-44Hz as in the analysis showed in Figure in the revised manuscript. (**B**) Statistics performed in a smaller time window (100-200ms) and only between 12-35Hz. Thus, we tried to exclude that the early significant cluster revealed by the original analysis is just spilling over from the larger and later β-rebound that would survive permutation-based statistics more easily than a smaller, separate cluster. And indeed, although we specifically tested around the outline of the early cluster in rIFG, it was still significant and not a spill-over from other clusters.

12. Frequency of interest – "Much of the discussion and the initial results regarding spectral power are focused on the broad β band from 12-30 Hz. Yet the analysis of Granger causality and DAI are focused on a slightly higher frequency range, 30-40 Hz, which is not usually considered as β when discussing the stopping nature of IFG (in fact, some would consider this in the range of low γ). How should one reconcile these different frequencies?a. Can the authors re-analyze in matched frequency bands and discuss the non – concordance in frequency given.

We agree with the reviewer that the parts of the TFR and connectivity (cGC) results were significant in different frequency bands. Yet, at the first step, both statistical analyses that aimed to test for differences between conditions, sSTOP and cAC-GO, had been actually performed in the same range, 8–44 Hz. Allow us to recapitulate how the analysis pipeline looked like for both methods before we try to reconcile the different outcomes.

**Author response image 4. sa2fig4:** Additional analysis of evoked responses and very low frequency oscillations. (**A**) Statistical comparison of evoked responses, sSTOP red curve, cAC-GO blue curve. A cluster-based permutation test was performed between 0 and 500 ms. Significant difference are highlighted by grey boxes. (**B**) Oscillator analysis including very low frequencies (2-44Hz). Here, the contrast of z-transformed sSTOP versus cCA-GO trials is show (Cluster-based permutations test, two-tailed, α_cluster_ = 0.5, n=59, significant clusters are outlined).

In case of the TFR analysis (Figure 3A in the original manuscript), the group statistics in the range 8–44 Hz revealed an early significant cluster in the rIFG (approximately in the range 10–34 Hz, peak at 25 Hz) and we supposed that this cluster could represent early processes of response inhibition. Since this cluster was spilling over to a lower frequency cluster, it was necessary to limit the definition of individual latency and power values to a frequency range in which not two obviously different neural effects take place simultaneously (i. e., α- and β-band oscillations). To draw a line, we decided to rely on the initial β band, 12–32 Hz, that was obtained by the sensor statistics during the tROI (100–350 ms) and that also was used for the source reconstruction (the question incorrectly stated a range from 12–30 Hz). Thus we could ensure that the TFR effect we analyzed matched best with the source reconstruction that revealed the rIFG and pre-SMA sources. Also note that in our reply to point 10 we showed that the β-band cluster was revealed independently of the lower-frequency cluster during the same time window.

In case of the connectivity analysis, we tested for differences in cGC, also within 8–44 Hz as for the TFR analysis at group level. Here, we found a broad cluster in the range 16–40 Hz with its peak between 24 to 28 Hz depending on the time window (at 28 Hz in the tROI). However, the cGC curves of both conditions were significantly different only between 30–40 Hz. Yet one should keep in mind two issues when interpreting these results. First, significance boundaries should not be overinterpreted. As pointed out by Sassenhagen and Draschkow (2019), frequency precision of statistical claims should not be overestimated when using cluster-based permutation tests. Given a higher statistical power, a wider range would probably be significant and hence, overlapping with the classical β-band range. Second, we don’t know exactly why the statistical results only reveal the right hand slope of the GC peak within the tROI. We guess that the data is just too noisy because of the limited amount of trials. Also note that the noise that comes with the GC analysis differs from the one of the TFR analysis. Additionally, as DAI is based on the estimated cGC values, the same considerations apply to the correlation between DAI and SSRT (also see b.).

In conclusion we would like to highlight the following aspects:

• The range of significant cGC differences was in fact overlapping with the cluster revealed by the TFR analysis, even if only in a small range between 30–32 Hz.

• Although the frequency band that showed significant differences between conditions (30-40 Hz) is not usually considered as β when discussing the stopping nature of IFG, the peaks obtained by both methods (28 Hz by cGC and 25 Hz by the TFR analysis) still could be assigned to the higher β band.

• The noise is not the same for the TFR and GC analysis – and this may differentially affect different frequencies.

• The constraints of cluster-based permutation tests should be considered when interpreting the significance of frequency boundaries.

Considering these facts, the slightly different spectral results of connectivity and TFR analysis can be reconciled. As a consequence, we interpret the effects revealed by both methods in a way that they are related to the same neural processes of response inhibition.

b. Moreover, the relation between DAI and SSRT is specific to 34 Hz. I understand that this has been corrected for multiple comparisons, but does this correction just account for the six discrete frequencies between 30-40 Hz, or does this account for the entire β band (or even all frequencies for that matter)? "

First, we identified differences between conditions in cGC between 8–44 Hz, using a cluster-based permutation test. This test revealed a significant frequency range between 30–40 Hz. Second, DAI was correlated with SSRT for each frequency in this significant range (30, 32, 34, 36, 38, 40 Hz). To identify whether a correlation was significant, a bootstrap method was applied, and the α threshold used for the CI was corrected by the number of the frequencies tested, i. e., corrected for six comparisons. Hence, since the correlation between DAI and SSRT was only tested for frequencies that were found to be significant at the first level, it did not seem appropriate to us to apply the correction for multiple comparison to the whole frequency range tested at the first level.

13. Frequency of interest – "This relates to interpretation – could the authors discuss and reference how an oscillation with a cycle length ~ 50ms could causally impact one region in the very short time frames demonstrated in this study (e.g SVM difference is 20ms in Figure 3C, ie less than half a cycle)."

We try to answer this question by illustrating a basic principle in physics. Allow us to use an example from simple electromagnetism and take a radio station that suddenly interrupts the playing of a song in order to announce breaking news. As soon as the sending radio station stops the song, this change has an instantaneous impact on the receiving device (only delayed by the propagation time through the medium, which is extremely short). The stopping of the song is not dependent on its frequency composition, i. e., low frequencies will stop at the same time as higher frequencies. Hence, the frequency of a process has nothing to do with the physical latency between sender and receiver. Likewise, the difference between two observed latencies is not limited to a cycle or multiples of a cycle.

As an additional demonstration, we simulated an auto-regressive model of a coupled system, where process A is oscillating with 20 Hz and process B with 30 Hz, and where A is driving B with a delay of 2 samples (corresponding to 16.67 ms with the sampling rate of 120 Hz used). When analyzing spectral Granger causality, we would expect no GC from A→A, B→A, and B→B. Only A→B should reveal a 20 Hz component. Indeed, this is what we obtain as results by our simulation (Author response image 5). Thus, it is well possible to detect Granger causality of signals with a period (50 ms) that is considerably larger than the coupling delay (16.67 ms).

**Author response image 5. sa2fig5:** Granger causality simulation. Simulation results of an auto-regressive model of a coupled system, where process A is oscillating with 20Hz and process B with 30 Hz, and where A is driving B with a delay of 16.67 ms. When analysing spectral Granger causality, we would expect no GC from A→A, B→A, and B→B. Only A→B should reveal a 20 Hz component. Indeed, this is what we obtained as results by our simulation. Thus, it is well possible to detect Granger causality of signal with a period (50 ms) that is considerably larger than the coupling delay (16.67 ms).

14. Granger causality – "Figure 4 – why does Granger causality go down with frequency? I would expect it to be flat except where there is a significant relation between signals. Could it be some kind of power confound or normalisation issue?"

We think that the GC does not go to zero because multiple frequencies contribute to the GC estimate, this makes the GC curves not flat outside the β-band peak. We additionally checked the influence of the estimation bias and ERF like frequency contributions. So far, they did not seem to have a substantial impact on the GC distribution.

Furthermore, there are many other MEG studies that show a similar GC distribution over frequency when GC estimates were computed in a similar manner. Please refer to e. g., Recasens et al. (2018) (Figure 5A, B), Rassi et al. (2019) (Figure 2C) or Seymour et al. (2019) (Figure 2).

However, we think that even if there were a potential disadvantage for higher frequencies (i.e., smaller GC values than for lower frequencies) due to normalization issues, this would not affect our conclusions because of two reasons: First and most importantly, we mainly interpret the GC results regarding the directionality. Second, we do not compare frequencies with each other within one GC analysis, but the difference between the conditions, sSTOP and cAC-GO. In conclusion, we don’t expect that the GC distribution as reported effects the interpretation of our results in any way.

15. DAI – "L. 475 – Was the correlation of DAI with SSRT independent of the IFG power correlation i.e. was the DAI regressor significant in the presence of rIFG power? If not, perhaps the DAI result could be explained by varying rIFG SNR that is known to confound directional measures."

Yes – an independent influence of DAI on SSRT had a probability of approximately 90% when controlling for rIFG power. This can be seen from our new Bayesian analysis that included rIFG power and DAI as additional regressors at the same time. For the results, please refer to our reply to question 1.

16. Clusters – "L. 1152 – the description of the cluster test seems to be inaccurate. If you took cluster sizes below 0.5th percentile of the distribution that would mean that you took very small clusters as well as very large. What usually happens is that people take large clusters for both positive and negative direction and apply Bonferroni correction for the two directions. Also why was a conservative threshold used p<0.01? Would there be additional clusters for p<0.05?"

We apologize for the incorrect description. The text was corrected accordingly: “Absolute cluster values above the 99.5th percentiles of the distribution of cluster sizes obtained for the permuted datasets were considered significant. This corresponds with an α threshold of _ = 0.01 that was Bonferroni-corrected for a two-tailed test.”

We used a more conservative threshold as we aimed to get crisp clusters and even more reliable results. This is possible due to the relatively large sample size used in our study. A threshold of *p* < 0.05 did not reveal additional clusters.

17. Code – "Although the authors included links to the data and the custom code in the cover letter, they do not appear in the paper itself. It would be helpful to include a supplementary table pointing to the toolboxes and custom functions used for each analysis. I was wondering particularly about cGC as no reference for the implementation was given and the conditional part sounds like more than is implemented in Fieldtrip."

We already provided a detailed list as *readme* file in the GitHub repository (https: //github.com/meglab/acSST). This list connects results (ordered by figure numbers) with code/scripts and data. However, we additionally created an overview box with all toolboxes used and the most important custom scripts (see Box 1 in the revised manuscript).

Conditional GC (cGC) is indeed implemented in FieldTrip, so we didn’t add an additional reference. Detailed script configurations can be found on our GitHub repository, too. For more information also consult the FieldTrip online documentation (https://www.fieldtriptoolbox.org/example/connectivity_conditional_granger/). The block approach we applied was cited. If you think a specific reference would be helpful in this context, please tell us and we’re happy to add it.

[Editors' note: further revisions were suggested prior to acceptance, as described below.]

A remaining issue relates to the new plots in Figure 3 and also the interpretation of the β signal. The new plots showing spectral activity per condition are welcome and confirm that there is indeed a reduced β desynchronization in the Stop condition. However, with the spectral (color) plots it is still difficult to accurately determine the time course of the β activity, which here is particularly important given the mechanistic interpretations of β and was a concern of reviewers.It would be advisable to include one further plot of a time series of the β filtered power in each condition separately as a separate time series. To put it another way – to show a line plot of the time resolved β power (rather than a spectrogram) with error bars. This would give a clearer view of whether the β plateaus, slows its desynchronization, slightly increases or whether the variance increases, all of which could be driving change in significance and are difficult to determine from the spectrogram. This is important for the mechanistic interpretation and particularly for the "Inhibitory role of β-band oscillations" Discussion section.

We thank you for suggesting this additional data representation. Indeed the color coding of the spectral plots is hard to interpret between the transition from negative to positive values. Hence, we averaged the β-band power within the frequency range used for the latency and power analyses, i. e., 12 to 32 Hz, based on *z*-transformed spectral data as used as input for the cluster-based permutation statistics in panels A–C. We added the results to Figure 3 as panel D. What we found was an increase of β power specifically for the rIFG in the sSTOP condition at the beginning of the temporal region of interest (100–350 ms) for the rIFG in the sSTOP condition, independent of a potential influence of the lower frequency cluster. To ensure that the shape of the averaged curve was not biased by lower frequency components, we also averaged for 14–32 Hz and 16–32 Hz, and indeed, the characteristics of the averaged curve particularly for the rIFG in the sSTOP condition did not change. Thus, there is evidence that there is indeed a termination of β-band desynchronization (BBD) in the rIFG in the sense that it was not just a slowing or plateauing of the BBD. We did not find an increase in synchronization strong enough such that β power was increased above baseline, but values reached baseline levels in rIFG. In this context, we would like to point out that there is big variation of β-band power changes across subjects. Some of them actually indeed showed a strong β *increase above baseline levels* before SSRT. Additionally, please note that the change of β-band power in the sSTOP condition in the rIFG is different from the one in pre-SMA and in the other areas (Figure 3, panel D). The rIFG is the only source revealing an early difference in the sSTOP condition compared to the AC-GO control condition. It is also particularly different from the premotor areas (rPMC, lPMC) that show the typical β-band modulation expected during planning and execution of a motor response.

While there is agreement that the study "suggests an active change of processing in rIFG that is related to stopping, while processing proceeds 'as usual' in cAC-GO trials", the role of β here and whether this it has a primary stopping function or these changes are secondary to a different mechanism for stopping in the IFG is still undetermined here. The current Figure 3 and your aligning with the status quo theory suggests that β changes here are more secondary and related to a withdrawal of an ongoing action initiation process rather than new active stopping process. It should be discussed further and potentially highlighted that a reduction of an ongoing motor initiation process (e.g. reducing β ERD) is different from a specific, distinct and active "stopping" process.Your current discussion was at times still felt to be implying that β was mechanistically a stopping signal (which would have been supported by an early increase in β with stopping) rather the possibility that β desync reflects motor initiation which was arrested (as implied by a slowing or plateauing or the ERD). An analogy might be the different between reducing pressure on an accelerator pedal, versus actively pressing a brake. In general this should be addressed throughout and the following are particular example areas of text which could be clarified to take account of this β discussion above:"We suggest that these differences then result in the observed net β-band power increase before SSRT". – This implies β increases, whereas the data thus far presented doesn't support this."Inhibitory role of β oscillations" – This isn't definitively shown here."Our results strongly support the interpretation that the rIFG acts as the initiator of response inhibition and that β-band oscillations play an important role for its neural implementation." – This implies β is mechanistic for response inhibition – but again this isn't shown here definitively, and β changes could secondarily result from a pausing of action initiation.

We agree that there are different theories on the nature of response inhibition, i. e., either as a pause of an already initiated go response, or as a distinct active stopping process (interpreted as a “brake”), as suggested by Aron et al. (2014). We now highlight this in the revised discussion. More importantly, we now clarify better that we did not provide evidence for a *causal* role of β-band changes during response inhibition. Hence, we considered our conclusions about β-band oscillations more carefully in the Discussion section. Yet, we would like to point your attention to the fact that we found significant Granger causality in time windows relevant for stopping, but not earlier, suggesting that active signaling from rIFG to pre-SMA specifically in the β band is an important part of the neural dynamics underlying stopping. In sum, the two findings, i. e., rIFG as source of stopping relevant information transfer in the β band and a termination of β-band desynchronization specifically in rIFG already before SSRT, suggest that response inhibition is implemented in part by β-band modulations. We hope that the revised interpretation in the Discussion section about the role of β-band oscillations has now gained in clarity with respect to how it is supported by our results.

Finally – whilst Figure 3 is a welcome addition – the curved greyed out area suggests a strong windowing effect from Time Frequency (TF) decomposition on short time window / epochs causing distortion at the edges which are worse for low frequencies (which have been masked out). This is unconventional to analyse and present like this, with normal practice being to perform the TF decomposition on the continuous data and to epoch power estimates afterwards, or to perform TF decomposition on a much wider epoch of data per trial and then re-epoch / chop down to the (same) time period of interest (e.g. -0.2 to 0.5). It is advised to repeat this analysis with better epoching – currently the windowing is artificially constraining the time and frequencies analysed.

We are aware that to avoid these masked-out areas, we would have to analyze longer time epochs. However, since our analysis was highly hypothesis driven and focused on β-band oscillations after the stop stimulus, we a priori defined the time period of interest between 100–350 ms after stimulus onset (see “tROI” in the manuscript) and pre-processed the raw data for this purpose. We additionally reserved 300 ms before and after this time period to perform a TF decomposition using at least three β-band cycles. More importantly, the LCMV beamformer used for source construction was most sensitive to source activity within the limited time window of interest. Hence, given these constraints, we indeed had to mask out the low frequency areas concerned to avoid boundary effects. Please note that for replying to questions 10 and 11 of the reviewers (before the first revision), we extended the time-frequency window from the original manuscript (0–500 ms, 8–44 Hz) to –200–500 ms, 2–44 Hz. We attach the figure with the extended time-frequency window (and masked-out areas) to this reply (Author response image 6) and use the figure with the originally analyzed shorter time windows in the resubmitted manuscript. We don’t see any crucial additional information for the reader by presenting wider epochs, and the results described in the manuscript are not referring to these extended time frequency windows. However, Figure 3 in the resubmitted manuscript still contains the additional plots with spectral activity *per condition*, which had definitely improved data interpretation compared to the original manuscript.

**Author response image 6. sa2fig6:** Β-band time-frequency representations of virtual channels at sources identified. First and second solid line indicate SSRT and SSRT_max_, first and second dotted line indicate 10% and 50%-percentiles of RT_AC-GO_ for selected AC-GO trials with RT_AC-GO_ > SSRT (RT _AC-GO_ is the duration between AC-GO signal and button, see Figure 1 of the manuscript). Time axis locked to STOP and AC-GO signal (0s). All plots shows the results of a cluster-based permutation test, two-tailed, α_cluster_ = 0.05, n=59, significant cluster are outlined. (**A**) Task-versus-baseline power for successful stop (sSTOP) trials. (**B**) Task-versus-baseline power for correct attentional capture go (cAC-GO trials). (**C**) Contrast of *z*-transformed successful stop (sSTOP) versus correct attentional capture go (cAC-GO) trials.